# Interplay Between Dysregulated Immune System and the Footprints of Blood-Borne miRNAs in Treatment Naive Crohn’s Disease and Ulcerative Colitis Patients

**DOI:** 10.3390/ijms262412042

**Published:** 2025-12-15

**Authors:** Emese Szilagyi-Tolnai, Anna Anita Szilagyi-Racz, Orsolya Kadenczki, Andras Balajthy, Peter David, Gabor Fidler, Peter Fauszt, Kristof Gal, Judit Remenyik, Karoly Palatka, Gyorgy Panyi, Melinda Paholcsek, Gabor Tajti

**Affiliations:** 1Faculty of Agricultural and Food Sciences and Environmental Management, Complex Systems and Microbiome-Innovations Center, University of Debrecen, 4032 Debrecen, Hungaryremenyik@agr.unideb.hu (J.R.); 2Department of Pediatrics, Faculty of Medicine, University of Debrecen, 4032 Debrecen, Hungarybalajthy.andras@med.unideb.hu (A.B.); 3Department of Oncoradiology, Faculty of Medicine, University of Debrecen, 4032 Debrecen, Hungary; 4Division of Gastroenterology, Department of Internal Medicine, University of Debrecen, 4032 Debrecen, Hungary; 5Department of Biophysics and Cell Biology, Faculty of Medicine, University of Debrecen, 4032 Debrecen, Hungary; panyi@med.unideb.hu; 6Institute of Physiology, Center for Physiology and Pharmacology, Medical University of Vienna, 1090 Vienna, Austria

**Keywords:** miRNA, inflammatory bowel disease, CD4^+^ T cells, differential diagnosis

## Abstract

Dysregulated T-cell-mediated immune responses are a hallmark of inflammatory bowel diseases (IBD), including Crohn’s disease (CD) and ulcerative colitis (UC). MicroRNAs (miRNAs) regulate various biological processes and play a significant role in the pathophysiology of numerous diseases. In this study, we aim to clarify the relationship between dysregulated immune response and altered miRNA signatures in patients with IBD. Our goal is to identify differentially expressed miRNAs that could potentially serve as diagnostic markers to differentiate between CD and UC. To quantify circulating miRNAs, we employed small RNA sequencing. To describe immune dysregulation, we determined the levels of circulating T-cell-related cytokines and the distribution of T-cell subpopulations in both circulation and in tissue samples. Our analysis revealed that 14 miRNAs exhibited significant expression differences between IBD patients and control subjects. These miRNAs may also implicate pathways associated with colitis-related colorectal carcinogenesis, suggesting their value in early risk assessment. Furthermore, we found that five miRNAs demonstrated a strong ability to discriminate between CD and UC patients. Additionally, levels of IL-22 and IFN-γ were significantly elevated in individuals with IBD. Notably, miRNA levels showed strong correlations with cytokine levels and T-cell subset distribution in both blood and tissue samples, exhibiting disease-specific patterns. In conclusion, we identified differentially expressed miRNAs in IBD patient groups, and a subset of these miRNAs might exhibit diagnostic potential to distinguish between CD and UC. Analyzing miRNAs in the blood of IBD patients may provide valuable insights into the underlying immune dysfunction.

## 1. Introduction

Inflammatory bowel diseases (IBD), such as Crohn’s disease (CD) and ulcerative colitis (UC), are complex disorders marked by persistent inflammation of the gastrointestinal tract [1,2,3]. IBD has become a significant global health issue, with increasing incidence rates observed in Western countries as well as in developing nations that have embraced a Western diet and lifestyle. In Europe, the prevalence of IBD is estimated to be as high as 0.5% [4].

Despite major advances in genetics, microbiome research, and immunology, the complex molecular cascades that connect barrier disruption, innate–adaptive immune crosstalk, and clinical heterogeneity are still incompletely understood [5,6]. In particular, there is a need for human studies that link immune cell phenotypes, cytokine networks, and post-transcriptional regulation in clearly defined patient populations.

Both CD and UC are chronic, progressive conditions characterized by the disruption of the intestinal barrier, leading to increased permeability and abnormal immune responses to environmental triggers [7]. In CD, inflammation occurs unevenly throughout the gastrointestinal tract and may affect any part of it [8]. In contrast, UC is limited to the colon, with inflammation affecting mainly the mucosa and submucosa. Both conditions could lead to recurrent flares and can also cause extraintestinal inflammation in organs such as the eyes or kidneys. Secondary complications, such as strictures and arthritis, can further diminish the quality of life for patients with IBD [9].

IBD-related inflammation is regulated by a complex interaction between innate and adaptive immune mechanisms. A variety of cytokines and chemokines orchestrate these processes in IBD, potentially disrupting the normal (physiological) inflammation in the gut [10].

T cell-dependent inflammatory response plays a crucial role in the pathogenesis. This is characterized by the upregulation of T-helper (Th)17 and pro-inflammatory Th1/Th2 pathways, alongside the downregulation of regulatory T-cell (Treg) pathways, which normally promote anti-inflammatory responses [11]. The circulating and local cytokine environment significantly influences T cell differentiation into functional subsets, including Th1, Th2, Treg, and Th17. This cytokine milieu also shapes the functionality of these adaptive immune cells, impacting disease manifestation [12].

In parallel, signaling axes such as NF-κB and JAK–STAT, together with pattern-recognition receptors (toll-like receptors (TLR), especially TLR4/TLR9), integrate signals from cytokines and the intestinal microbiota [13,14,15]. Microbiota-derived metabolites, bacterial products, and barrier damage continuously modulate these pathways. This links environmental triggers, genetic risk loci, and immune activation in IBD [16].

MicroRNAs (miRNAs) are increasingly recognized as regulators of various cellular processes, including inflammation, cell differentiation, and immune responses [17]. There is significant interest in understanding the biological functions of miRNAs and their altered expression patterns in different pathological conditions [18].

However, most human studies that have profiled miRNAs relied solely on mucosal biopsies or examined cohorts that were under various disease-related treatments (i.e., were not therapy naïve). These factors complicate mechanistic interpretation and biomarker translation. Likewise, cytokine signatures and T cell subset distributions have typically been investigated separately, without parallel miRNA profiling that could reveal coordinated, pathway-level regulation. As a result, the miRNA–cytokine–T-cell axis remains only partially characterized in IBD, and its potential relevance for early colorectal cancer (CRC) risk in chronic colitis is inferred rather than directly examined [19,20]. With this, there is a clear unmet need for data that simultaneously capture the miRNA–cytokine–T-cell axis in strictly therapy-naïve, newly diagnosed patients with CD and UC. Moreover, there remains a gap in studies linking these molecular axes to colitis-associated CRC risk, such as by connecting candidate miRNAs to JAK–STAT/NF-κB signaling, epithelial barrier regulation, and known CRC-related miRNA networks.

In IBD, dysregulated miRNA expression can affect genes linked to immune cell responses, the integrity of the epithelial barrier, tissue homeostasis, and the delicate balance between pro-inflammatory and anti-inflammatory signaling. More than 100 different miRNAs have been identified in innate and adaptive immune cells, playing a key role in regulating molecular pathways that control development and function [21]. Additionally, the regulation of protein expression by miRNAs is essential for T cell activation, proliferation, and cytokine production [22].

Several canonical IBD-associated miRNAs, such as miR-155 and miR-223, have been shown to fine-tune these pathways. miR-155 promotes Th1/Th17 responses and NF-κB activation, whereas miR-223 modulates myeloid cell function, NLRP3 inflammasome activity, and epithelial responses at the mucosa–microbiota interface [23,24]. Together with miR-21, miR-146a, and members of the miR-17~92 family, these miRNAs orchestrate JAK–STAT and TLR-driven cytokine networks, highlighting that the miRNome is tightly embedded in microbiota–immune–epithelium cross-talk [25].

In IBD, particularly UC, dysregulated miRNA expression, chronic inflammation, and ongoing activation of the immune system are key factors in the development of colitis-associated colorectal cancer (CAC) [26,27]. The NF-κB and STAT3 pathways contribute to a tumor-promoting inflammatory environment by enhancing cell proliferation and survival while inhibiting apoptosis. Additionally, inflammation-related oxidative stress further increases genomic instability [24,25]. Here, dysregulated miRNAs intersect with oncogenic and tumor-suppressive pathways. MiR-21 acts as an oncomiR by engaging PI3K/AKT and STAT3 signaling, miR-145 exerts tumor-suppressive effects by constraining migration and epithelial–mesenchymal transition, while miR-106a influences cell growth, apoptosis, and autophagy [28,29,30]. Furthermore, miR-155 and miR-223 contribute to CAC pathogenesis by promoting NF-κB/JAK–STAT signaling, modulating myeloid-derived suppressor cells, and reshaping the mucosal cytokine milieu [31,32]. Common mechanisms include disruption of the epithelial barrier, changes in the cytokine network (such as TNF-α and IL-6), TLR signaling (TLR4 and TLR9), and miRNA-mediated gene regulation. Integrative assessment of these pathways and miRNA panels holds promise for risk stratification, early detection, and targeted intervention [33,34]. These observations support the concept of a molecular continuum linking chronic intestinal inflammation, miRNA dysregulation, and malignant transformation.

Identifying the interactions between cytokine signatures, T cell subset profiles, and miRNA expression is crucial for understanding disease mechanisms and may also guide therapeutic interventions. Additionally, dysregulated and differentially expressed miRNAs may serve as valuable biomarkers for disease diagnosis, prognosis, and monitoring of treatment responses.

There is a clear unmet need for integrative studies that (i) profile the circulating miRNome, (ii) quantify key pro- and anti-inflammatory cytokines, and (iii) define CD4^+^ T-cell subset distributions in both the circulation and inflamed mucosa within a single, well-characterized, therapy-naïve IBD cohort. Such a systems-level approach is essential not only for refining disease mechanisms but also for identifying composite biomarker signatures that may aid in the differential diagnosis of CD versus UC and, potentially, in early risk stratification for CRC-related complications. In our exploratory study, we aimed to address these gaps by combining small RNA sequencing, targeted miRNA validation, comprehensive cytokine profiling, and detailed CD4^+^ T-cell subset analysis in treatment-naïve patients with CD and UC, as well as healthy controls. Specifically, we aimed to (i) define distinct circulating miRNome signatures in newly diagnosed, therapy-naïve IBD; (ii) assess whether differentially expressed miRNAs can serve as potential diagnostic biomarkers to distinguish CD from UC; (iii) characterize pro- and anti-inflammatory cytokine profiles and CD4^+^ T-cell subsets (Th1, Th2, Th17, Treg) in blood and inflamed tissue; (iv) explore how cytokine patterns associate with miRNA expression; and (v) determine how miRNAs and cytokines relate to the distribution of T cell subsets both in the circulation and at sites of inflammation. By explicitly mapping our validated miRNA panel onto IBD- and CRC-associated pathways, we sought to position the miRNA–cytokine–T-cell axis as a potential mechanistic bridge between immune dysregulation, barrier failure, and early CRC-related changes in IBD, while simultaneously evaluating its translational utility as a minimally invasive biomarker framework.

Here, we combine unbiased small RNA sequencing with targeted miRNA validation, multiplex cytokine profiling, and parallel characterization of circulating and mucosal CD4^+^ T-cell subsets in a strictly therapy-naïve IBD cohort. This integrated design enables us to delineate disease-specific miRNA–cytokine–T-cell interaction patterns, identify circulating miRNA signatures that discriminate CD from UC, and anchor these signatures to pathways implicated in colitis-associated colorectal cancer. Together, our findings position the miRNA–cytokine–T-cell axis as a mechanistically informative and potentially clinically actionable framework for biomarker development in newly diagnosed IBD. The novelty of our research is that we examined miRNA expression in a group of therapy-naive patients and compared the results with the characteristics of dysregulated immune responses, thus identifying early, treatment-free molecular abnormalities of the disease.

## 2. Results

### 2.1. Characteristics of the Study Cohort

In this prospective study, we recruited 27 participants from the University of Debrecen, Faculty of Medicine, specifically at the Departments of Gastroenterology and Pediatrics, between 2018 and 2021 (Table 1). The cohort included 8 individuals with CD, 10 with UC, and 9 healthy controls. The baseline demographics and clinical characteristics of the participants were balanced in terms of age: those with UC (31.1 ± 5.85 years), those with CD (24.5 ± 10.8 years), and the healthy controls (30.9 ± 5.67 years). The cohort was also balanced by sex, consisting of 15 males and 12 females.

### 2.2. General Description of Sequencing Results

A total of 1,340,406 ± 1,068,598 cDNA reads (each 75 base pairs long) were generated per sample, resulting in 37,889,412 reads for each cDNA library. The majority of the sequenced reads ranged from 21 to 23 nucleotides in length. After filtering out low-quality tags, removing adaptors, and eliminating contaminants, more than 90% of the clean reads were retained. We analyzed the types of small RNA sequences based on their uniqueness and length distribution. In summary, over 95% (± 2%) of the clean reads were classified as miRNAs.

### 2.3. Small Noncoding RNA Transcriptome Quantitative Analysis Unveiled Common and Distinctive miRNAs in the Study Cohort

We used high-throughput small RNA sequencing to analyze the miRNA expression profile of our study cohort, which included 27 participants: 8 with Crohn’s disease, 10 with ulcerative colitis, and 9 healthy volunteers. A total of 2477 miRNAs were identified, but 2013 of them were excluded from the analysis due to a low read per million (RPM < 10) across all samples. This left us with 464 miRNAs that had an RPM of 10 or greater in at least one of the samples examined. We focused our further analyses on this subset of miRNAs (see Appendix A). The number of uniquely expressed and shared miRNAs is illustrated in a Venn diagram (Figure 1A). In total, 457 miRNAs were consistently detected across all experimental groups.

Furthermore, we identified three miRNAs—hsa-miR-10395-3p, hsa-miR-1248, and hsa-miR-195-3p—that were expressed only in patients with IBD, as shown in Table 2. In addition, hsa-miR-6837-3p, hsa-miR-1246, and hsa-miR-374c-5p were detected exclusively in the UC and healthy control groups, whereas hsa-miR-6509-3p was uniquely identified in both the healthy control and CD groups.

Circo plots displaying the distribution profiles of the core 464 miRNAs (Figure 1B) indicate significant expression changes when comparing healthy controls to either CD or UC patients. The RPM values of all 464 miRNAs in all samples are listed in Appendix A.

### 2.4. Differentially Expressed miRNAs Were Revealed Between IBD Patients and Healthy Groups

Differential expression analysis was conducted to identify miRNAs that were distinctly expressed between the experimental groups (Figure 2A). In this analysis, samples from patients with CD and UC were pooled together to form the IBD group. The volcano plot revealed 141 miRNAs exhibiting statistically significant expression differences between the control and the IBD group. Among these, 120 miRNAs showed reduced expression, while 21 miRNAs demonstrated increased expression compared to the control group. Based on these findings, we focused our further analyses on the 141 differentially expressed miRNAs (Appendix A).

We performed ordination pattern analysis on the differentially expressed miRNAs to identify distinct clusters within our experimental groups (Figure 2B,C). The beta diversity relationships were summarized in two-dimensional scatterplots, where each point represents a patient and the distances between points indicate differences in miRNA expression. Distance-based dissimilarity matrices revealed that IBD had a significant influence on overall community variations, resulting in non-overlapping clusters (Cluster 1 and Cluster 2) (Figure 2B). However, when we further divided the IBD group into CD and UC subgroups, no significant impact on β-diversity profiles was observed (Figure 2C).

To determine if any miRNAs had the potential to distinguish between the CD and UC groups, we evaluated the corresponding raw *p*-values from the volcano analysis (Figure 2D). To manage the large number of miRNAs to be tested, we employed receiver operating characteristic (ROC) analyses on the significantly altered miRNAs. The adjusted *p*-values, along with the area under the ROC curve (AUC) values for all miRNAs, are detailed in Appendix A. Overall, 123 miRNAs were found to have an AUC of 0.8 or higher.

Based on a literature search for altered expression levels reported in previous studies related to the intestinal system (including IBD, irritable bowel syndrome, and colon cancer), we selected 62 miRNAs for further analysis [35,36,37,38,39,40,41,42,43,44,45,46,47,48,49,50,51,52,53,54,55,56,57,58,59,60,61,62,63,64,65,66,67,68,69,70,71,72,73,74,75,76,77,78,79,80,81,82,83,84,85,86,87,88,89,90]. A cluster heat map was created to represent the expression levels of the selected 62 miRNAs (Figure 2D). Among these, 12 miRNAs were upregulated, and 50 were downregulated in the CD and UC groups compared to healthy controls. The expression patterns of these miRNAs were also clustered to support the possible diagnostic potential of circulating miRNA signatures in IBD. We identified two main clusters separating healthy controls from diseased patients, but no distinct clustering emerged between CD and UC (Figure 2D). For PCR validation, we selected 62 miRNAs (Figure 2D) based on our small RNA sequencing results and previously published studies relevant to intestinal biology.

### 2.5. Validation of Differentially Expressed miRNAs by RT-qPCR

To confirm the consistent gene expression tendencies seen in our miRNA-seq data, 62 significantly altered miRNAs with high potential diagnostic value were tested via RT-qPCR as well (Figure 3B) (Appendix A).

According to our findings, 14 miRNAs were validated successfully by RT-qPCR across our sample groups. Of note, these expression patterns were consistent with the RNA sequencing results (Figure 3A) (Appendix A).

We found 11 miRNAs (hsa-miR-145-5p, hsa-miR-191-5p, hsa-miR-101-3p, hsa-miR-106b-5p, hsa-miR-19b-3p, hsa-miR-20a-5p, hsa-miR-15a-5p, hsa-miR-17-3p, hsa-miR-103a-3p, hsa-miR-454-3p, hsa-miR-16-5p) that were significantly downregulated, while three miRNAs (hsa-miR-151a-3p, hsa-miR-16-2-3p, hsa-miR-106a-5p) were up-regulated compared to our control samples.

It is important to highlight that five miRNAs (hsa-miR-145-5p, hsa-miR-20a-5p, hsa-miR-103a-3p, hsa-miR-106a-5p, and hsa-miR-191-5p) showed significantly different expression levels between the CD and UC groups according to our RT-qPCR validation.

Receiver operating characteristic (ROC) curve analysis was performed to evaluate the ability of these miRNAs to distinguish between patients with CD and those with UC (Appendix A). Three of the five miRNAs (hsa-miR-106a-5p, hsa-miR-103a-3p, and hsa-miR-191-5p) demonstrated AUC = 1. Moreover, hsa-miR-145-5p, with AUC = 0.97, and hsa-miR-20a-5p with AUC = 0.77 also exhibit reasonably high AUC values in our dataset. It is important to highlight that these are only preliminary conclusions due to the small sample size. Further validations in larger, independent cohorts are needed to strengthen our observations.

### 2.6. Association of Circulating Cytokine Levels with Circulating miRNAs in IBD 

The association between miRNA expression and standard laboratory parameters was assessed in IBD patient groups. We first evaluated the distribution of circulating white blood cells and C-reactive protein levels (Figure 4A). We found a significantly higher percentage of lymphocytes in the UC group (24.40 ± 9.33) compared to the CD group (14.64 ± 5.83), with a *p*-value of less than 0.005. In contrast, serum CRP levels were significantly higher (*p* < 0.05) in the CD group (62.44 ± 71.19) compared to the UC group (3.74 ± 3.12), which is consistent with the existing literature.

To determine the differences in circulating cytokine levels between controls and IBD patients, T cell-related cytokines were measured (Figure 4B). IFN-γ [CD:41.02 ± 22.79, UC:119.28 ± 97.68, control 51.65 ± 30] and IL-22 [CD:5.42 ± 3.37, UC:5.81 ± 3.35, control 0.38 ± 0.85] were significantly upregulated in both CD and UC compared to healthy controls. Of note, no significant differences were observed between the CD and UC groups in any of the cytokines examined.

We conducted Spearman correlation analyses of the 14 miRNAs and cytokine levels to evaluate how this bidirectional system is dysregulated in Crohn’s disease and ulcerative colitis (Figure 4C). We observed notable differences in the correlation patterns between miRNAs and cytokine profiles in the patient groups with CD and UC. We found that IL-5, IL-13, IL-17A, IL-4, IL-21 and IL-22 were negatively correlated with hsa-miR-454-3p, hsa-miR-19b-3p, hsa-miR-106a-5p, hsa-miR-191-5p, hsa-miR-103a-3p in both CD and UC patients.

We also observed a disease-specific correlation pattern between miRNAs and cytokine profiles in the patient groups with CD and UC. In the UC group, IL-2 has a positive correlation with hsa-miR-151a-3p, with a correlation coefficient (r value of 0.52). Conversely, in the CD group, a strong negative association was observed between IL-2 and hsa-miR-16-2-3p, with a value of −0.92.

IL-10 exhibited a positive correlation with several miRNAs in UC: miR-151a-3p (r value: 0.80), hsa-miR-16-2-3p (r value: 0.89), hsa-miR-15a-5p (r value: 0.57), hsa-miR-17-3p (r value: 0.63). In contrast, only a weak positive correlation was observed between IL-10 and hsa-mir-15a-5p (r = 0.43) in CD.

In CD, strong negative correlations were found between TNF-α and various miRNAs: hsa-miR-454-3p (r value: −0.76), hsa-miR-19b-3p (r value: −0.70), hsa-miR-20a-5p (r value: −0.73), hsa-miR-106a-5p (r value: −0.70), hsa-miR-191-5p (r value: −0.73) and hsa-miR-103a-3p (r value: −0.84). In contrast, only weak correlations were recorded for UC.

Interestingly, hsa-miR-151a-3p showed a negative correlation (r value: −0.61) with IL-6 in CD, while it displayed a strong positive association (r value: 0.89) in UC. Additionally, INF-γ had a negative correlation (r value: −0.71) with hsa-miR-151a-3p in CD patients, whereas a weak positive correlation (r value: 0.28) was noted in UC.

A similar correlation pattern was observed in both CD and UC groups regarding IL-5 and several microRNAs. The correlations were found as follows: hsa-miR-454-3p (CD: r value: −0.52, UC: r value: −0.67), hsa-miR-106a-5p (CD: r value: −0.54, UC: r value: −0.60), hsa-miR-191-5p (CD: r value: −0.53, UC: r value: −0.61) and hsa-miR-103a-3p (CD: r value: −0.41, UC: r value: −0.67). IL-13 also showed a negative correlation with specific microRNAs in both UC and CD groups: hsa-miR-106a-5p (CD: r value: −0.53, UC: r value: −0.68), hsa-miR-191-5p (CD: r value: −0.49, UC: r value: −0.65) and hsa-miR-103a-3p (CD: r value: −0.23, UC: r value: −0.67).

Notably, IL-21 was positively correlated with hsa-miR-16-2-3p (r = 0.55) and hsa-miR-16-5p (r = 0.56). However, a negative correlation was found with the following miRNAs in CD: hsa-miR-101-3p (r = −0.67), miR-17-3p (r = −0.66), hsa-miR-454-3p (r = −0.53), hsa-miR-19b-3p (r = −0.60), hsa-miR-20a-5p (r = −0.56), hsa-miR-106a-5p (r = −0.59) and hsa-miR-191-5p (r = −0.56) in CD. Only weak positive correlations were observed between IL-21 and microRNAs in UC.

### 2.7. Correlation Between CD4^+^ T Cell Subtype Distribution, Cytokine Levels, and microRNAs in IBD

Flow cytometry analysis was conducted to assess the distribution of CD4^+^ T cell subtypes in patients with CD or UC. The representative gating strategy is illustrated in Figure 5A. Distribution of T-helper (Th1, Th2, Th17) and the T-regulatory (Treg) cells in the circulation, as well as in tissue biopsy samples, was evaluated (shown as percentage of CD4^+^ cells) (Figure 5B,C). There were no significant differences in the circulating CD4^+^ T cell population frequencies between the CD (Th1: 5.38% ± 1.2), (Th2: 10.33% ± 2.11), (Th17: 27.06% ± 9.13), (Treg: 1.26% ± 0.54) and UC (Th1: 6.25% ± 2.65), (Th2: 11.0% ± 5.71), (Th17: 32.44% ± 16.40), (Treg: 2.12% ± 1.59) groups (Figure 5B).

Intestinal biopsy samples taken from spots showing macroscopic signs of active inflammation (inflamed) and macroscopically non-inflamed spots (non-inflamed) derived from 4 to 5 patients with CD and 8–10 patients with UC were also characterized (Figure 5C). The percentage of Th1 and Th2 was similar in both active (Th1: 13.96% ± 2.52, Th2: 9.11% ± 0.77) and non-inflamed (Th1: 14.29% ± 5.37, Th2: 7.25% ± 2.86) sites of CD patients. The percentage of T helper cells (Th1, Th2, Th17) showed moderate yet not significant differences between inflamed (Th1: 12.81% ± 3.74), (Th2: 13.09% ± 6.13), (Th17: 23.13% ± 11.49) and non-inflamed (Th1: 22.13% ± 12.76), (Th2: 11.58% ± 5.11), (Th17: 25.04% ± 12.98) samples of UC patients. Analyses of regulatory T cells revealed significant differences between inflamed (6.42% ± 2.75) and non-inflamed (3.31% ± 1.89) tissue sites of UC patients. Furthermore, the percentage of Treg cells was notably higher in UC (6.42% ± 2.75) than in CD (2.36% ± 1.25) study population at the active inflammatory site.

Spearman correlation analyses were implemented to assess the association patterns between CD4^+^ T cell subset frequencies, expression of circulating miRNAs, and circulating cytokine levels in CD and UC patients (Figure 5D,E). In the CD patient group, Treg cells showed negative association both in circulation and inflamed biopsy samples with hsa-miR-454-3p (r values: −0.66 and −0.62, respectively), hsa-miR-19b-3p (r values: −0.66 and −0.62 respectively), hsa-miR-106a-5p (r values: −0.66 and −0.62 respectively), hsa-miR-20a-5p (r values: −0.66 and −0.62 respectively), hsa-miR-191-5p (r values: −0.66 and −0.62 respectively), hsa-miR-103a-3p (r values: −0.66 and −0.62 respectively), hsa-miR-16-2-3p (r values: −0.66 and −0.62 respectively) and hsa-miR-151a-3p (r values: −0.66 and −0.62 respectively) (Figure 5D). Notably, comparison of circulating and active inflammatory tissue of CD patients showed the same relationship pattern between CD4^+^ T cell subset frequencies and miRNA levels.

Circulatory CD4^+^ T cell subset frequencies showed negative correlation with hsa-miR-16-2-3p for Th2 (r value: −0.79) as well as Th17 (r value: −0.68) subsets. Moreover, in the inflamed tissue of CD patients, we observed the same association pattern between hsa-miR-16-2-3p and Th17 (r value: −0.79) and also in non-inflamed tissue with Th1 (r value: −0.72). Strong positive correlation was observed between Th2 with hsa-mir-151a-3p (r value: 0.8) and Th17 with hsa-miR-151a-3p (r value: 0.74) and IL-22 (r value: 0.74) in the blood samples from CD patients. Furthermore, IL-22 levels positively correlated with Th2 and Treg frequencies (r values: 0.70 and 0.69, respectively) in inflamed tissue samples of CD patients and also with Th1, Th2, and Treg frequencies in non-inflamed biopsy samples (r values: 0.72, 0.70, and 0.68, respectively).

In CD patients, the percentage of circulatory Th17 positively correlated with IL-17 and IL-22 cytokines, whereas negatively associated with IL-21. In the case of UC patients, negative correlations were observed between Th17 with IL-17, IL-22, and IL-21 cytokines.

It is important to note that correlations were found to be stronger between CD4^+^ T cell subtypes with circulating miRNAs in both blood and tissue biopsy samples than that of circulating cytokines.

In UC patients, substantially fewer parameters exhibited strong correlations (Figure 5E). In blood samples of UC patients, the percentage of Th1 cells showed a positive correlation with IL-5, IL-13, and IL-2 (r values: 0.71, 0.74, and 0.69, respectively), while the percentage of Th2 cells positively correlated with IL-10 (r value: 0.67) and negatively correlated with hsa-miR-16-2-3p (r value: −0.68). The miRNA hsa-miR-16-2-3p also exhibited a negative correlation with Treg cell frequencies in blood samples of UC patients (r value: −0.77). In inflamed tissue biopsy samples of UC patients, only hsa-mir-16-2-3p showed strong correlation with Th2, Th17, and Treg (r values: −0.88, −0.94, and –0.81, respectively).

### 2.8. Discriminatory Power of Biological Parameters Between CD and UC Study Groups

Random forest analysis was performed to assess the differential diagnostic potential of the investigated parameters in distinguishing CD and UC (Figure 6A). The partial dependence plots indicate that parameters located to the right of zero are more prevalent in UC patients, while those on the left are more common in CD patients. The size of each dot in the plot reflects the discriminatory power of the parameters. Since the distribution of different T cell subsets was only characterized in biopsy samples from five CD patients and nine UC patients, only the data from these patients were included in our analyses.

Our results indicate that CRP, hsa-miR-191-5p, hsa-miR-103a-3p, hsa-miR-106a-5p, hsa-miR-145-5p, and IL-4 are the most effective markers for distinguishing between the CD and UC groups.

Following our random forest analyses, we conducted ordination pattern analyses (Figure 6B) to identify distinct clusters among our experimental groups. We assessed standard laboratory blood parameters, 14 significantly altered miRNAs (qPCR data), circulating cytokine levels, and the frequencies of CD4^+^ T cell subsets in both inflamed tissue samples and circulation. Beta diversity relationships were summarized using multidimensional scaling (MDS) scatterplots, where each point represents an individual patient. The distance-based dissimilarity matrices revealed two non-overlapping clusters: one for CD and one for UC, each representing different spatial ordinations.

## 3. Discussion

Inflammatory bowel diseases, including Crohn’s disease and ulcerative colitis, are a group of chronic conditions marked by increased inflammation of the gastrointestinal tract, usually characterized by a relapsing and remitting clinical course [6].

Despite extensive research, the pathogenesis of IBD remains incompletely understood [5]. However, it is well-established that many factors significantly contribute to the disease pathology [91]. Numerous studies underscore the importance of miRNAs in various diseases, including IBD [19,92,93]. Our exploratory study examines the relationship between miRNA expression profiles and parameters that describe the status and immune response in treatment-naive IBD patients.

We conducted transcriptome sequencing on blood samples from patients with Crohn’s disease, ulcerative colitis, and healthy volunteers. We observed differences in the expression profiles of miRNAs between IBD patients and healthy controls. In total, we identified 141 miRNAs that were significantly altered when comparing IBD patients to healthy controls, with only 23 miRNAs showing increased expression in the IBD group. Notably, several miRNAs that exhibited decreased expression in IBD patients have been postulated to play an important role in maintaining immune homeostasis (hsa-miR-122, hsa-miR-1146a, hsa-miR-495) [18], supporting the intestinal mucus barrier (hsa-miR-191, hsa-miR-146a, hsa-miR-106a) [94], regulating T cell differentiation (hsa-miR-21, hsa-miR-106b) [95], apoptosis (hsa-miR-148a) [96] and autophagy (hsa-miR-106b) [95]. The decreased level of these miRNAs found in IBD patients may support their possible role in the disease pathology by contributing to the dysregulation of the aforementioned mechanisms.

We found that three miRNAs—hsa-miR-10395-3p, hsa-miR-1248, and hsa-miR-195-3p—were only expressed in patients with IBD. In addition, hsa-miR-6837-3p, hsa-miR-1246, and hsa-miR-374c-5p were detected in the UC and healthy control groups, while hsa-miR-6509-3p was expressed in both the healthy control and CD patients. Interestingly, none of these miRNAs showed statistically significant expression differences between our study groups. According to a previous study, hsa-miR-1248 has been suggested to function as a tumor-suppressive miRNA in colorectal cancer by targeting and inhibiting PSMD10, indicating a potential regulatory role in colorectal tumorigenesis [97]. Furthermore, hsa-miR-195-3p has been reported to function as a tumor-suppressive miRNA in hepatocellular carcinoma, where its reduced expression contributes to UBE2I overexpression and promotes cancer cell migration and invasion [98]. hsa-miR-10395-3p, identified in peritoneal lavage-derived exosomes, is associated with peritoneal metastasis in gastric cancer and appears to contribute to tumor metastasis and chemoresistance [99].

We identified 14 miRNAs (hsa-miR-145-5p, hsa-miR-191-5p, hsa-miR-101-3p, hsa-miR-106b-5p, hsa-miR-19b-3p, hsa-miR-20a-5p, hsa-miR-15a-5p, hsa-miR-17-3p, hsa-miR-103a-3p, hsa-miR-454-3p, hsa-miR-16-5p, hsa-miR-151a-5p, hsa-miR-16-2-3p, hsa-miR-106a-5p) that exhibited statistically different expression in IBD patients compared to the control group. All of these miRNAs have been described in previous studies of UC and CD [100]. miR-145 is widely downregulated in CRC, and in IBD, where its loss aligns with enhanced NF-κB signaling, epithelial–mesenchymal transition (EMT), and tumor progression [101]. In CRC cells, reduced miR-145-5p derepresses TWIST1 and EMT programs, boosting proliferation, migration, and invasion, and permits increased MMP-9 expression [102]. Thus, miR-145-5p loss not only mirrors inflammation but may foster a pro-tumor milieu and more aggressive, metastasis-prone phenotypes [103].

A previous report indicated that hsa-miR-15a-5p is upregulated in UC patients and negatively regulates epithelial junctions by modulating cdc42 [64]. In contrast, we observed decreased expression levels of hsa-miR-15a-5p in both CD and UC patient groups. This discrepancy may reflect differences in sample type (peripheral blood vs. mucosal biopsies), disease stage, or the strictly therapy-naïve status of our cohort, and underlines the need to interpret miRNA directionality within the context of a clinical setting and tissue compartment.

miR-106a-5p was one of the first miRNAs described to exhibit altered expression in IBD patients. miR-106a-5p is a member of the conserved miR-17 family, sharing a similar seed sequence [39]. Research has shown that the miR-17 family targets multiple genes related to autophagy, including ATG16L1, which has been suggested to play a significant role in CD [104]. Our findings indicate that hsa-miR-106a-5p is expressed at higher levels in CD patients, which aligns with previously published studies [22].

Elevated levels of hsa-miR-101-3p result in reduced proteasome levels, consequently leading to cell cycle arrest and apoptosis via p53 and cyclin-dependent kinase inhibitor 88 accumulation. Conversely, the loss of hsa-miR-101-3p leads to pathological cellular processes such as tumorigenesis, altered metabolism, and prevention of apoptosis [59,105]. Here, we found decreased levels of hsa-miR-101-3p in both CD and UC groups, which might indicate its involvement in IBD pathology through abnormal regulation of apoptosis and metabolism.

Furthermore, hsa-miR-103a-3p and hsa-miR-191-5p have been associated with epithelial integrity and metabolic stress responses, raising the possibility that their up- or downregulation reflects early epithelial barrier disruption in newly diagnosed IBD [106].

Importantly, many of the validated miRNAs in our panel, including miR-106a, miR-20a, miR-145, miR-101, and miR-16, have already been implicated in IBD and/or CRC. Thus, the novelty of our work does not lie in discovering entirely new miRNA species, but rather in: (i) confirming and refining these signatures in a well-defined, strictly therapy-naïve cohort, and (ii) embedding them within a systems-level framework that jointly considers cytokine networks and T cell subset distributions.

While key inflammatory miRNAs such as miR-155 and miR-223 did not emerge among our top differentially expressed, validated candidates, they are well-established regulators of JAK–STAT signaling, NF-κB activation, and microbiota-driven inflammation. Their absence from our final panel may again be linked to blood-based sampling and cohort size; nevertheless, our data complement these canonical mediators by highlighting a partially overlapping but distinct set of circulating miRNAs that track with systemic immune dysregulation.

Innate and adaptive immune responses play crucial roles in the pathogenesis of IBD [107]. Activated dendritic cells (DCs) and macrophages, key components of the innate immune system, interact with various subsets of T cells, which are part of the adaptive immune response. This interaction occurs through different cytokines that regulate the inflammatory response in UC and CD. Cytokines are also involved in disrupting the normal state of controlled inflammation [10]. The local and systemic cytokine environment influences the levels of reactive oxygen species, nitric oxide, leukotrienes, and prostaglandins, factors that may contribute to the dysregulation of the inflammatory response in affected tissues [10]. Clearly, an imbalance in these cytokines contributes to the pathogenesis of IBD [108]. It is important to note that previous studies have shown that cytokine signatures and the resulting immune responses differ significantly among the various forms of IBD [2]. Therefore, assessing cytokine signatures is essential for understanding disease pathology and guiding therapeutic strategies.

Motivated by these findings, we assessed the circulating cytokine profile in plasma samples from patients and control individuals. We found significantly increased levels of IL-22 in both CD and UC patients compared to the controls. A substantial body of literature suggests that IL-22 may have protective effects in IBD, such as promotion of wound healing and tissue regeneration. However, it is important to note that excessive IL-22 expression can also exacerbate inflammation [109]. In our study, while IL-22 levels were elevated in comparison to healthy controls, we did not observe any differences between CD and UC patient groups.

Consistent with previous studies [108,110], we found significantly higher levels of IFN-γ in the CD and UC groups compared to the control samples. IFN-γ is a proinflammatory cytokine that stimulates macrophages and neutrophils, promoting immune-cell recruitment by inducing the expression of adhesion molecules on epithelial cells [108,110].

The IL-10 knockout mouse model is a well-established tool to study IBD. In this model, the absence of IL-10 leads to increased production of IFN-γ and the development of spontaneous colitis [111,112]. IL-10 is an anti-inflammatory cytokine that plays a vital role in reducing mucosal inflammation by inhibiting antigen presentation and the subsequent release of pro-inflammatory cytokines. Previous studies have reported low levels of IL-10 in the inflamed tissues of CD patients [113]. However, in our study, circulating IL-10 levels were not significantly altered in the CD and UC patient groups compared to controls.

Increased production of TNF-α is a hallmark of IBD [94]. TNF-α serves as a key pro-inflammatory cytokine with widespread effects [114]. Serum levels of TNF-α correlate with clinical and laboratory indicators of disease activity in both CD and UC, rendering it a primary target for biological therapies [107]. In our study, we observed slightly elevated levels of TNF-α in the circulation of CD and UC patients compared to controls; however, the differences did not reach statistical significance.

According to previous findings, hsa-miR-106a-5p, members of the miR-17~92 family, have been noticed in regulating Th1/Th17 differentiation and modulating key cytokines such as IL-6 and IL-23, suggesting that their altered expression may contribute to early immune activation [115].

T cell-mediated immune responses play a crucial role in the pathogenesis of IBD [116]. Therefore, modulating T cell function is an important strategy to treat IBD, as underlined by several new biological treatment options. These include therapies that target T cell homing, such as vedolizumab [117], and those that target T cell-related cytokines, such as ustekinumab [118]. In this study, we analyzed the distribution of Th1, Th2, Th17, and Treg CD4^+^ cells in blood and biopsy samples from patients with CD and UC. In blood samples, no significant differences were found between CD4^+^ subset distribution upon comparing the CD and UC groups. Our results indicated that the proportion of Th17 cells in the circulation of patients with CD positively correlated with the level of IL-22 (r value: 0.59), but negatively correlated with IL-21 (r value: −0.50). Interestingly, in patients with UC, the proportion of circulating Th17 cells was negatively associated with the levels of IL-17, IL-22, and IL-21 (r value: 0.51) cytokines. These findings align with previous studies showing that Th17 cells were more abundant in the peripheral blood of IBD patients, and significant increases in Th17 cytokines (IL-17, IL-21, and IL-23) were noted in the inflamed mucosa of these individuals [119].

CD is traditionally considered a disease characterized by an exaggerated Th1- and Th17-mediated immune response, with elevated production of key cytokines such as IL-12, IL-23, IFN-γ, and IL-17. Furthermore, Th1 cells can help to mitigate intestinal inflammation by secreting IL-10 and IL-2, which stimulate Treg differentiation. Prior studies suggest that Th1 cells significantly contribute to the development of IBD, particularly in CD [12]. We observed positive correlations between circulating Th1 cells and INF-γ and IL-21 (r = 0.41 and r = 0.40, respectively) in the CD patient group. However, no correlations were found between the percentages of inflamed tissue Th1 cells and INF-γ or IL-21.

In contrast to CD, UC is considered a Th2 (and Th9)-driven disease, associated with elevated levels of cytokines such as IL-13, IL-5, and IL-9. In UC patients, we found that the percentage of Th1 cells in blood positively correlated with IL-5 (r value: 0.71), IL-13 (r value: 0.74), and IL-2 (r value: 0.69). Circulating Th2 cell proportions positively correlated with IL-10 and negatively correlated with hsa-miR-16-2-3p (r value: −0.57). Additionally, circulating Treg proportions were negatively associated with hsa-miR-16-2-3p (r value: −0.77). In inflamed tissue samples, we observed strong negative correlations between miR-16-2-3p and Th2 (r value: −0.88) and Treg (r value: −0.81) cell proportions, while a strong positive correlation was noted between miR-16-2-3p and Th17 cell (r value: 0.94) proportions.

MicroRNAs, which are important regulators of protein expression, play an important role in T cell physiology—including development, activation, and proliferation [78]. Therefore, through regulating T cell function, miRNAs also significantly impact cytokine production and inflammation [120]. So far, only a limited number of studies have explored the relationship between miRNA profiles, cytokine levels, and the presence of both circulatory and tissue-resident CD4^+^ T cells in patients with IBD. In summary, most of our findings align with existing published data, which is detailed below.

The loss of Treg suppressive function mediated by TNF-α is associated with the induction of a specific miRNA, miR-106a, in both humans and mice [121]. This microRNA, in turn, suppresses the release of IL-10 by modulating the NF-κB promoter region. We found that hsa-miR-106a has only a weak positive correlation (r value: 0.32) with IL-10 in UC, while no significant associations were found in the CD patient group. Additionally, a strong negative correlation (r value: −0.62) was observed between Treg levels and hsa-miR-106 miRNA in the CD study group, which aligns with previous findings.

According to literature, miR-19 promotes the production of Th2 cytokines and enhances inflammatory signaling by targeting the phosphatase and tensin homolog (PTEN) gene, the signaling inhibitor of suppressor of cytokine signaling 1 (SOCS1), and the deubiquitinase A1 [122]. In our study, miR-19b-3p exhibited a negative correlation with Th2 cytokines, specifically IL-5, IL-13, and IL-4, both in CD and UC patient groups. However, we did not observe any correlation between miR-19b-3p levels and proportions of Th2 cells in either the circulation or the inflamed mucosa of patients with CD and UC.

In experimental autoimmune encephalitis (EAE), a non-intestinal autoimmune disease, miR-20 has been shown to inhibit Th17 cell differentiation and reduce IL-17 cytokine levels [123]. We observed strong negative correlations between miR-20a-5p and IL-17 levels (r value: −0.76) and a weak negative correlation between miR-20a-5p and the circulating Th17 (r value: −0.23) ratios in patients with CD. These findings are consistent with the phenomena observed in EAE.

Altered expression of hsa-miR-16 has been observed in patients with UC. As noted by Tian and colleagues, miR-16 facilitates the nuclear translocation of NF-κB p65 protein, leading to the increased expression of pro-inflammatory cytokines such as IFN-γ and IL-8 in gut epithelial cells [124]. We found a positive correlation between levels of hsa-miR-16 and IFN-γ (r value: 0.40), as well as a strong correlation with the percentage of Th17 cells (r value: 0.94) in the inflamed tissue sample of UC patients.

Hsa-miR-151a-3p has previously been shown to prevent LPS-induced IL-6 production by targeting STAT3 [125,126]. Our results indicate that hsa-miR-151-3p exhibits a strong negative correlation with IL-6 in both CD and UC patient groups. Additionally, we observed a strong positive correlation between the percentages of circulating Th2 and Th17 (r values: 0.80 and 0.74, respectively) cells as well as IL-17F (r value: 0.83) in CD patients.

Hsa-miR-145 may play a role in controlling inflammatory responses by targeting IKKα, IKKβ, IL1R1, GP130, and TNFR2, leading to a decrease in NF-κB pathway activation [13]. Correspondingly, we found that levels of hsa-miR-145-5p were significantly lower in IBD patients compared to control groups (*p* < 0.005). Furthermore, hsa-miR-145-5p was differentially expressed between CD and UC study groups (*p* < 0.005). We also found strong positive correlations between hsa-miR-145-5p and IL-13 (r value: 0.77) in CD patients. Notably, positive correlations were also observed between hsa-miR-145-5p and the percentages of circulatory Th2 and Th17 cells (r values: 0.57 and 0.55, respectively) in the CD patient group.

Overall, we found that miRNAs with altered expression in IBD correlate with T cell subset distribution as well as disease-specific cytokine levels, suggesting that an altered miRNA expression profile may contribute to the disease pathogenesis and reflect the underlying immune dysregulation in IBD. At the same time, many of the key miRNAs we report (miR-106a, miR-145, miR-20a, miR-16, miR-101) are also known players in CRC biology, acting on apoptosis, autophagy, epithelial plasticity, and JAK–STAT/NF-κB pathways. Thus, the miRNA–cytokine–T-cell axis characterized here may not only capture inflammatory dynamics but may also touch upon early molecular events that predispose to colitis-associated CRC, complementing prior mucosa-focused studies [9]. The gold standard for the diagnosis of IBD relies on clinical manifestations, endoscopy, and histopathological analyses [127]. However, there is an urgent need to identify non-invasive and potentially cost-effective alternative methods for the diagnosis and monitoring of disease activity in IBD [128]. To date, numerous studies have sought a non-invasive biomarker that can accurately differentiate between CD and UC [2].

In IBD, inflammation is primarily driven by three key pro-inflammatory cytokines: IL-6, TNF-α, and IL-1β [129]. These cytokines trigger the production of CRP, an acute-phase protein produced by hepatocytes, which serves as a surrogate marker of inflammatory processes throughout the body. Our findings indicated that CRP levels were significantly higher in CD compared to those with UC (*p* < 0.05), which aligns with previous reports [130]. Although CRP levels differ between the two patient groups, its lack of specificity and the fact that it can be upregulated in numerous other diseases limit its utility as a biomarker for IBD diagnosis.

The remarkable stability of miRNAs in circulation and throughout the body allows researchers to utilize them as biomarkers, potentially also for IBD [131]. In our study, we investigated whether specific miRNAs that differ significantly between CD and UC patient groups could aid in differential diagnosis. Our results showed that five miRNAs—hsa-miR-145-5p, hsa-miR-20a-5p, hsa-miR-103a-3p, hsa-miR-106a-5p, and hsa-miR-191-5p- exhibited significantly different expression levels when comparing CD and UC patients. Among these, hsa-miR-20a-5p, hsa-miR-103a-3p, and hsa-miR-106a-5p have previously been identified as potential biomarkers in studies examining miRNA expression in CD and UC patients [128]. We observed that all five miRNAs (hsa-miR-106a-5p, hsa-miR-103a-3p and hsa-miR-191-5p, hsa-miR-145-5p, hsa-miR-20a-5p) demonstrated promising discriminatory power (AUC = 0.77–1). However, due to the small sample size, further validations with a larger independent cohort are needed to draw far-reaching conclusions about whether these miRNAs have any real diagnostic power to discriminate between CD and UC.

To promote this, we conducted a Random Forest analysis to evaluate the power of these parameters in distinguishing between CD and UC patient groups. Our findings revealed that CRP, hsa-miR-191-5p, hsa-miR-103a-3p, hsa-miR-106a-5p, hsa-miR-145-5p, and IL-4 had the highest potential to discriminate between CD and UC in our cohort. In conclusion, all five differentially expressed miRNAs—hsa-miR-145-5p, hsa-miR-20a-5p, hsa-miR-103a-3p, hsa-miR-106a-5p, and hsa-miR-191-5p—appear to be potentially useful both individually and in combination to support the differential diagnosis of CD and UC. Our preliminary data suggest that hsa-miR-106a-5p, hsa-miR-103a-3p, hsa-miR-191-5p, hsa-miR-145-5p, and hsa-miR-20a-5p may serve as potential biomarkers for IBD, with the ability to discriminate ulcerative colitis from Crohn’s disease. Nevertheless, larger cohorts and additional validation analyses will be required to confirm their diagnostic utility.

Taken together, our analyses extend previous reports by showing that, in a strictly therapy-naïve cohort, a limited circulating miRNA panel combined with CRP and selected cytokines can achieve excellent separation between CD and UC, positioning this exploratory work as translational between mechanistic immunology and clinically actionable biomarker combinations.

Our study provides insights into the potential applicability of miRNAs as differential diagnostic markers of IBD, which, together with parameters routinely measured in clinical laboratories, may pave the way to the future development of non-invasive and cost-effective diagnostic and therapy monitoring tools for CD and UC. Our results highlighted that dysregulated miRNA expression may have a role in shaping systemic cytokine levels and may also affect the function and distribution of T cells, possibly contributing to IBD pathology. By explicitly relating our validated miRNA panel to established IBD- and CRC-associated miRNAs (including miR-155 and miR-223) and to key pathways such as NF-κB and JAK–STAT, we position our findings within the current literature and emphasize that the miRNA–cytokine–T-cell axis described here is a plausible mechanistic link between inflammation control, immune dysregulation, and early CRC-related changes in IBD.

A key strength of our study is that all recruited patients were strictly therapy-naive at the time of sample collection, meaning they had not received any IBD-related medication prior to the sampling. However, this strict inclusion criterion also represents a significant limitation, as it resulted in a limited sample size and a relatively long sample collection period. Limited sample size also prevented us from implementing in-depth patient stratification (i.e., based on sex, age, or disease severity). Increasing sample size and extending the cohort would therefore be crucial to underline our findings. A small sample size might also have resulted, that only five miRNAs were found to exhibit differential expression between CD and UC disease groups. Additionally, because miRNAs have a pleiotropic effect, they can influence various biological pathways [117]. This complexity makes it challenging to assert a causal relationship between miRNAs to IBD pathology and to support their use in differential diagnostics. Ongoing research and validation are essential to translate these findings into clinical practice.

The exclusive enrolment of therapy-naïve patients is advantageous in terms of minimizing pharmacological confounding factors, which enhances the mechanistic interpretability of the miRNA–cytokine–T-cell relationship; it inevitably resulted in a relatively modest, single-center cohort. This likely reduces statistical power and contributes to the emergence of only a limited set of robustly discriminatory miRNAs between Crohn’s disease and ulcerative colitis. A further constraint arises from the fact that miRNA profiling was confined to peripheral blood owing to limited biopsy material, precluding direct alignment of circulating and mucosal miRNA landscapes and their spatial relationship to local immune and epithelial compartments. Parallel small RNA sequencing of blood and intestinal biopsies, ideally coupled with spatially resolved immune phenotyping, will be required to clarify tissue origin, compartmentalization, and the translational robustness of the candidate miRNAs.

In addition, the cross-sectional and predominantly correlative nature of the study does not permit firm causal inference or characterization of temporal dynamics across flare, remission, and treatment initiation. Longitudinal sampling from diagnosis through defined therapeutic milestones, combined with time-series analyses, should help determine whether the identified miRNA–cytokine–T-cell patterns track with disease course, therapeutic response, and trajectories towards dysplasia or colorectal cancer.

Finally, neither microbiome nor host genetic data were integrated, and functional perturbation of individual miRNAs or their targets was beyond the scope of this work. Future studies that embed the miRNA–cytokine–T-cell axis within a broader multi-omic framework and incorporate targeted gain- and loss-of-function experiments should be crucial to move from association towards causality and to underpin the development of clinically applicable, biologically grounded biomarker panels.

## 4. Materials and Methods

### 4.1. Patient Population, Study Group, and Biological Samples

IBD patients (8 with CD and 10 with UC) were prospectively recruited at the University of Debrecen, Faculty of Medicine, Department of Gastroenterology, or at the Department of Pediatrics. The diagnosis of IBD was based on clinical manifestations and was proven by endoscopic examination followed by histological characterization. The severity of diseases was evaluated using the CD Activity Index (CDAI) in case of the adult CD patients, while the Mayo score was used for adult UC patients. For pediatric patients, the Pediatric Crohn’s Disease Activity Index (PCDAI) and Pediatric Ulcerative Colitis Activity Index (PUCAI) were used to score CD and UC severity, respectively. All the recruited patients were therapy naïve at the time of the visit (i.e., were not receiving any treatment for IBD), where the clinical data, laboratory parameters, as well as biological samples (blood samples, tissue biopsy samples of bowel sections having macroscopic signs of active inflammation present (inflamed) or absent (non-inflamed)) were taken. Blood samples obtained from healthy volunteers matched for age and sex (*n* = 9) served as control samples. The pediatric patients recruited were all adolescents (age: 12–18 years), whereas the adult patients were relatively young (age: 19–45 years). Given the therapy naïve nature of recruited patients, the exact onset of disease is unknown. Driven by the relatively uniform age distribution, unknown disease onset, and low patient count, data were only stratified based on primary diagnosis (CD or UC).

### 4.2. Blood and Tissue Sample Preparation

Blood samples of IBD patients and controls were drawn to either heparin-anticoagulated Vacutainer^®^ or Vacutainer^®^ CPT™ Mononuclear Cell Preparation Tubes (both from Becton Dickinson, Franklin Lakes, NJ, USA), and peripheral blood mononuclear cells (PBMC) were separated either using standard density gradient centrifugation or using the CPT™ tubes according to the manufacturer’s instructions. PBMCs were immediately used for subsequent immune cell characterization.

Tissue biopsy samples, taken from inflamed and non-inflamed bowel sections were immediately stored in HBSS (with calcium, magnesium and 10 mM HEPES) at 4 °C. Samples were transferred and processed immediately by enzymatic digestion using 500 µg/mL Collagenase IV (Thermo Fisher Scientific, Waltham, MA, USA) and 250 U/mL DNAse I (AppliChem GmbH, Darmstadt, Germany) at 37 °C for 1h with continuous gentle agitation. Enzymatic digestion was supported by mechanical trituration after 30 min, using 19 G blunt-ended needles and 5 mL syringes. Subsequently, samples were filtered using cell strainers with 70 µm-pore size, to obtain single cell suspensions, which were centrifuged and washed 3 times with HBSS (without calcium, magnesium, and HEPES) before immune cell characterization.

### 4.3. Pro- and Anti-Inflammatory Cytokine Measurement

The levels of circulating cytokines were measured from plasma samples of patients or healthy controls. To this, blood samples were drawn into EDTA-anticoagulated BD Vacutainer^®^ tubes (Becton Dickinson, Franklin Lakes, NJ, USA), which were centrifuged at 1000× *g*, 10 min at RT to acquire blood plasma. Samples were immediately transferred to −80 °C until the measurements. Cytokines, characteristic of the function of different T cell subsets (either pro- or anti-inflammatory), were measured by cytokine bead array, using the LEGENDPlex Human T Helper Cytokine Panel (BioLegend, San Diego, CA, USA) according to the manufacturer’s instructions with respect to origin-specific sample preparation, calibration, measurement, as well as data evaluation. Samples were measured using NovoCyte 3000 (RYB) flow cytometer equipped with an automated sample loader (Agilent Technologies, Santa Clara, CA, USA), and the acquired data were evaluated using the software provided with the cytokine bead array kit (LEGENDPlex Software v8.0, BioLegend).

### 4.4. Determination of T Cell Subpopulations

The relative frequency of T cell subpopulations (Th1, Th2, Th17, and Treg) in PBMC as well as samples of tissue biopsy origin were determined via flow cytometry using a BD FACSAria™III cytometer (BD Biosciences, San Jose, CA, USA).

After sample preparation as described before, cells were stained using FITC anti-human CD4 mAb (clone: SK3), Brilliant Violet 421™ anti-human CCR6 mAb (clone: G034E3), PE anti-human CCR4 mAb (clone: L291H4), APC anti-human CXCR3 mAb (clone: G025H7), PerCP/Cyanine5.5 anti-human CD25 mAb (clone: BC96) and PE/Cyanine7 anti-human CD127 mAb (clone: A019D5), all from BioLegend. Using proper single-stained samples, spillover compensation was implemented online, as well as during offline data evaluation using Flowjo 10 software (BD Biosciences, San Jose, CA, USA). After gating for CD4^+^, T cell subsets were identified as Th1 (CXCR3+, CCR4− CCR6−), Th2 (CCR3−, CCR4+, CCR6−), Th17 (CCR4+, CCR6+) and Treg (CD25+, CD127−). Missing values are due to missing or low-quality samples, which were excluded from the analyses.

### 4.5. RNA Extraction

Whole blood samples collected into EDTA-containing tubes were used for microRNA analyses. To avoid environmental contamination, isolation was carried out in a class II laminar-flow cabinet. Total RNA was extracted from whole blood using the MagMax mirVana total RNA isolation kit according to the manufacturer’s instructions (Thermo Fisher Scientific, Waltham, MA, USA). Furthermore, negative isolation control (NIC) of sterile nuclease-free water was also prepared along with the samples.

RNA was quantified using the Qubit miRNA Assay Kit and Qubit™ 4 Fluorometer (Thermo Fisher Scientific, Waltham, MA, USA). RNA integrity was assessed using the Agilent 4200 TapeStation System with RNA ScreenTape and RNA ScreenTape Buffer (Agilent Technologies, Santa Clara, CA, USA). Only samples with RNA integrity number (RIN) values greater than 6 were used for downstream analysis. RNA purity was verified by spectrophotometry using the NanoDrop™ 2000 Spectrophotometer (Thermo Fisher Scientific, Wilmington, DE, USA). High-quality RNA samples were stored at −80 °C until library preparation.

### 4.6. Library Preparation and Sequencing

Small RNA libraries were prepared using a NEBNext^®^ Small RNA Library Prep Set kit for Illumina^®^ (New England Biolabs Inc., Ipswich, MA, USA) in accordance with the manufacturer’s instructions. Briefly, 500 ng of total RNA (in 6 μL volume) was subjected to multiplex adapter ligation (using 3′ and 5′ SR adaptors), reverse transcription, primer hybridization, and PCR amplification. Amplified cDNA fragments were purified using the QIAQuick PCR Purification Kit (Qiagen, Hilden, Germany) and MagSI-NGS^PREP^ Plus beads (Magtivio B.V., Nuth, The Netherlands). Size selection was performed on E-Gel^®^ EX 2% Agarose (Thermo Fisher Scientific, Waltham, MA, USA) with the E-Gel™ Power Snap Electrophoresis Device (Thermo Fisher Scientific Baltics UAB, Vilnius, Lithuania) according to the manufacturer’s guidelines.

Library size distribution and integrity were verified with the Agilent 4200 TapeStation System using D1000 ScreenTape (Agilent Technologies, Waldbronn, Germany) and D1000 Sample Buffer (Agilent Technologies, Waldbronn, Germany). Equimolar pooling of each library was performed to generate a final pooled library. The concentration of the libraries was adjusted to 4 nM using 10 mM Tris (pH 8.5). The library pool was denatured with 0.2 N NaOH. 1% PhiX Control Library (Illumina, San Diego, CA, USA) was also denatured and used as an internal sequencing control. Sequencing was carried out on an Illumina NextSeq 550 System (Illumina, Singapore), producing 75-nucleotide-long single-end reads at an average depth of 3.5 million reads per sample.

### 4.7. Bioinformatic Analyses

Two sequencing runs were performed, with samples randomly distributed across batches. Both sequencing runs included samples from all study groups to minimize potential batch effects and ensure balanced representation during downstream analyses.

Demultiplexed reads were processed using the Cutadapt software (v. 4.0) to remove the adaptor sequence (AGATCGGAAGAGCACACGTCTGAACTCCAGTCAC), and low-quality reads [132]. Read quality metrics were evaluated with the FastQC program version 0.12.0 (Babraham Institute, Cambridge, UK). Additional trimming was performed with the Trimmomatic program (v. 0.40) [133].

Clean reads were further processed using the miRge 2.0 software [134]. This pipeline provides annotation of microRNA-seq data against a known miRNA database. In brief, the software splits the reads into two groups based on a predetermined threshold for the length of the target reads, set at 28 bases, then maps the reads to miRBase (v. 22) [135] and calculates microRNA abundance. Then, we analyzed read counts using the edgeR R package (v. 4.9.1) [136]. Normalization was performed using the trimmed mean of M values (TMM) method to account for sequencing depth and compositional variance. Normalized expression levels were expressed as reads per million (RPM), adjusted by the TMM-derived scaling factors as follows:$$RPM=\frac{read\;counts\;per\;microRNA}{total\;read\;count}\times \frac{1,000,000}{TMM\;factor}$$

miRNAs below 10 RPM were excluded from further analyses. Differences between groups were assessed using the nonparametric Kruskal–Wallis test, followed by Dunn’s post hoc test for pairwise comparisons. Volcano plots were generated using the Enhanced Volcano R package (v. 1.11.3). For confirmatory statistics, raw read counts were converted to reads per million (RPM) to account for library size differences, and nonparametric Kruskal–Wallis tests were performed across clinical subgroups. Hierarchical clustering heatmap was generated using the pheatmap R package (version 1.1.12).

Venn diagrams were assessed by using the limma R package (version 3.28.14) 50. Spearman correlation coefficients were calculated, and correlation plots were constructed using the corrplot R package (version 0.92). To evaluate the differences in miRNA expression between groups, principal coordinates analysis (PCoA) was performed using the vegan v.2.6-4 package in R.

The diagnostic potential of candidate miRNAs was determined using easyROC (ver. 1.3.1) [137]. Receiver Operating Characteristic (ROC) curves were generated by plotting sensitivity (true positive rates) against the 1-specificity (false positive rates). The area under the ROC curve (AUC) was calculated to quantify the diagnostic performance of the miRNAs.

Optimal cut-off points for individual miRNAs were determined using the Youden Index (J = sensitivity + specificity −1), identifying the threshold that maximized overall diagnostic accuracy [138]. To assess cut-off stability, 95% confidence intervals were computed.

To provide robust estimates of model performance, a stratified k-fold cross-validation procedure was implemented [139]. The full dataset was randomly divided into k equal-sized folds, each containing proportional representation from all outcome groups. In every iteration, one-fold was held out for testing, while the remaining k− folds were used for model training and cut-off determination. Each fold served once as the test set. Sensitivity, specificity, and AUC were calculated for each iteration, and the mean AUC and corresponding 95% confidence interval were calculated across all folds, providing a robust estimate of model accuracy for selected miRNA biomarkers.

To identify key predictors that could effectively differentiate between CD and UC patient groups, we employed a machine learning strategy that utilizes the Random Forest (RF) approach, as described by Breiman [140]. Partial Dependence Plot (PDP) and multidimensional scaling (MDS) plots were generated using the RandomForest package v.4.7-1.1. in R. The RF model was built using the log2-normalized expression levels of miRNAs, CRP, cytokine levels, as well as T-cell subset percentages from five CD and nine UC patients as input features. A total of 10,000 trees (n_tree_ = 10,000) were generated to ensure model stability and reliable predictions. The dataset was randomly divided into training (80%) and test (20%) subsets. Variable importance was assessed using the MeanDecreaseGini and MeanDecreaseAccuracy coefficients to identify biomarkers distinguishing between CD and UC patients. The feasibility of building a classification model was further evaluated by generating an MDS plot, which mapped the coordinates derived from the proximity matrix of the RF analysis, where distance represents dissimilarity. Clusters of points in the plot indicate samples that the RF model considers similar.

### 4.8. Validation of miRNA-Seq Data by Real-Time q-PCR

Validation of miRNA sequencing results was performed using real-time quantitative PCR (RT-qPCR). Total RNA (2 ng per reaction) was subjected to miRNA-specific reverse transcription using the TaqMan™ Advanced miRNA cDNA Synthesis Kit (Thermo Fisher Scientific, Waltham, MA, USA), following the manufacturer’s instructions. RT-qPCR for 62 selected miRNAs was carried out using TaqMan™ Gene Expression Assays (Thermo Fisher Scientific, Waltham, MA, USA) on a LightCycler^®^ 480 Real-Time PCR System (Roche Diagnostics, Risch-Rotkreuz, Switzerland) (Appendix A). All measurements were conducted in triplicate, including negative controls (no template controls (NTCs) replacing the DNA template with nuclease-free water. The expression level of hsa-let-7i-5p was the most stable among all the samples; therefore, it was used as an endogenous control for normalization. Mean cycle threshold (Ct) values of technical replicates were used for gene expression quantification. Relative gene expression of miRNAs was calculated by the comparative ΔΔCt method according to Livak’s formula. After normalization to the housekeeping miRNA (hsa-miR-let-7i-5p) using the ΔCt method, relative expression levels were calculated using the ΔΔCt method, with the control group (healthy volunteers) serving as the calibrator. Based on the results of small RNA sequencing, we selected a subset of 62 miRNAs for further validation. Selection criteria included the magnitude of differential expression between IBD patients and healthy controls, as well as prior evidence from the literature implicating these miRNAs in intestinal physiology, immune regulation, or inflammatory bowel disease pathogenesis.

### 4.9. Statistical Analysis

Statistical analyses were conducted using GraphPad Prism version 8.0 (GraphPad Software, La Jolla, CA, USA). Statistical comparison among multiple groups was performed with the Kruskal–Wallis test, and intergroup differences were tested with Dunn’s post hoc test. The level of significance was set at *p*-value < 0.05. Levels of significance were shown as: * *p* ≤ 0.05; ** *p* ≤ 0.005; and *** *p* ≤ 0.0005. Numerical data are presented as mean ± standard deviation (SD).

## Figures and Tables

**Figure 1 ijms-26-12042-f001:**
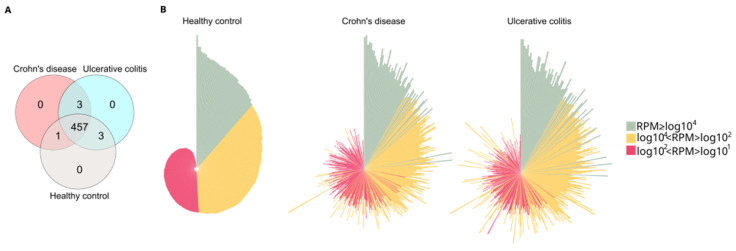
Analysis of core and distinctive miRNAs in the study cohort. (**A**) Venn diagram illustrating the distribution of sequenced miRNA across different study groups. The diagram shows the number of unique microRNAs in healthy control, Crohn’s disease (CD) and Ulcerative colitis (UC) groups. The overlap regions indicate the number of microRNAs shared across two or more study groups. (**B**) Normalized distribution patterns of the 464 miRNAs in the healthy control, CD, and UC groups are depicted using circular bar plots. In these circo plots, green, yellow, and red represent miRNAs with high (RPM ≥ log10^4^), medium (log10^4^ < RPM < log10^2^), and low (log10^2^ < RPM < log10^1^) RPM values, respectively. The order of miRNAs (represented by the bars) was consistent across all groups, and the lengths of the bars correspond to the log10 of RPM. RPM (reads per million).

**Figure 2 ijms-26-12042-f002:**
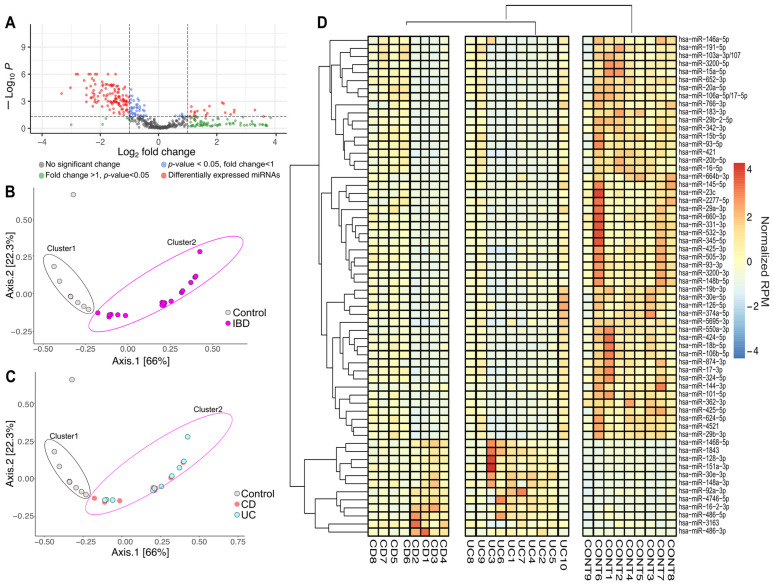
Altered miRNA expression patterns between inflammatory bowel disease (IBD) and healthy controls. (**A**) A volcano plot was created to identify miRNAs with differential expression between the IBD cohort and the healthy control group. Statistically significant results (*p* < 0.05) along with fold changes greater than 2 or less than −2 were determined using the Linear Models for Microarray Data (LIMMA) statistical model. The horizontal dashed line indicates the -log_10_ transformed *p*-value threshold (*p* < 0.05), while the vertical dashed line represents the fold change thresholds of log_2_(fold change) = 1 and −1, corresponding to 2-fold up- or downregulation, respectively. Each point represents an individual miRNA, with red circles showing miRNAs with statistically significant differential expression (|log_2_FC| > 1; *p* < 0.05). (**B**) The multidimensional scaling (MDS) plot shows differences in miRNA distribution patterns between individuals with inflammatory bowel disease (IBD; pink) and healthy controls (gray). (**C**) MDS analysis further stratified by IBD subtype, displaying the distribution of Crohn’s disease (CD; red) and ulcerative colitis (UC; blue) patients relative to healthy controls (gray). (**D**) Hierarchical cluster heatmap of significantly different miRNAs (*n* = 62) showed distinct clusters. Expression levels are presented as normalized RPM values relative to the mean across all samples, with red indicating upregulation, while blue indicates downregulation. IBD (inflammatory bowel disease, CD and UC groups pooled), CD (Crohn’s disease), UC (Ulcerative colitis), CONT (Control group), RPM (reads per million).

**Figure 3 ijms-26-12042-f003:**
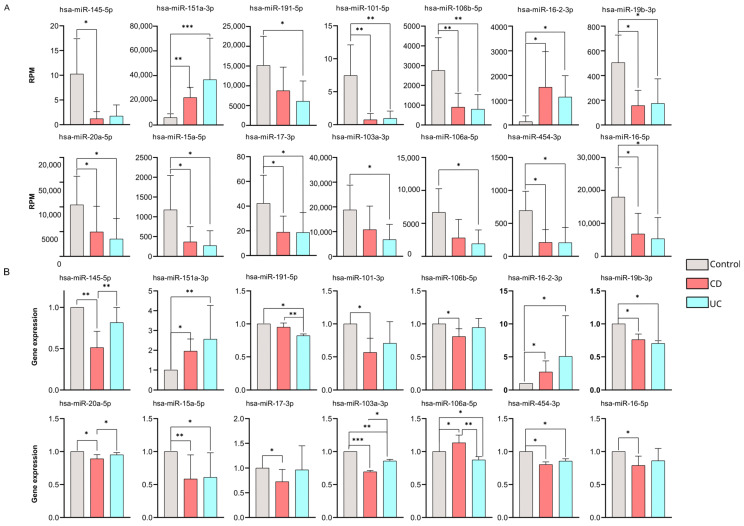
Validation of differentially expressed miRNAs by RT-qPCR. (**A**) Bar charts represent normalized RPM values for CD (red) and UC (blue) patients and health controls (gray). (**B**) The expression changes in the 14 microRNAs are validated by RT-qPCR. Bar charts represent the relative gene expression of each microRNA, calculated using Livak’s formula and normalized to hsa-let-7i-5p. Data are shown as mean ± SD. Significant differences were determined by the Kruskal–Wallis test with Dunn’s multiple-comparison. Asterisks report statistical significance * *p* < 0.05; ** *p* < 0.005; *** *p* < 0.001. CD (Crohn’s disease), UC (Ulcerative colitis), RPM (reads per million).

**Figure 4 ijms-26-12042-f004:**
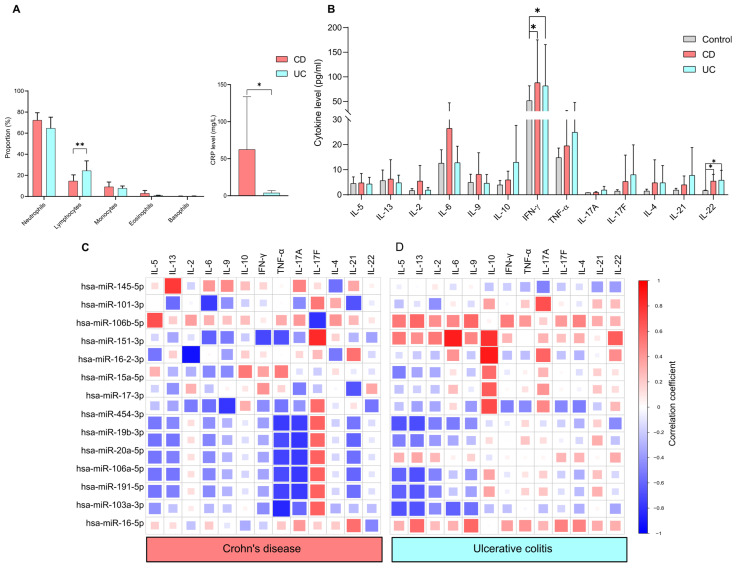
Correlation between circulating cytokine levels and miRNAs in IBD. (**A**) Bar charts show the distribution of white blood cell counts and C-reactive protein levels in CD (red) and UC (blue) patients. (**B**) Circulating cytokines, measured by cytokine bead array in CD (red) and UC (blue) patient groups relative to healthy controls (gray). Spearman correlation coefficients identified associations between miRNAs and cytokine levels in (**C**) CD and (**D**) UC groups. Based on the RT-qPCR validation, normalized expression levels of 14 miRNAs were used for the analyses. Correlation coefficients (−1 to +1) indicate positive (R ≥ 0; red) or negative (R < 0; blue) association. Data are shown as mean ± SD. Statistical significance was calculated using the Kruskal–Wallis test with Dunn’s multiple-comparison. Asterisks report statistical significance * *p* < 0.05; ** *p* < 0.005. CD (Crohn’s disease), UC (Ulcerative colitis).

**Figure 5 ijms-26-12042-f005:**
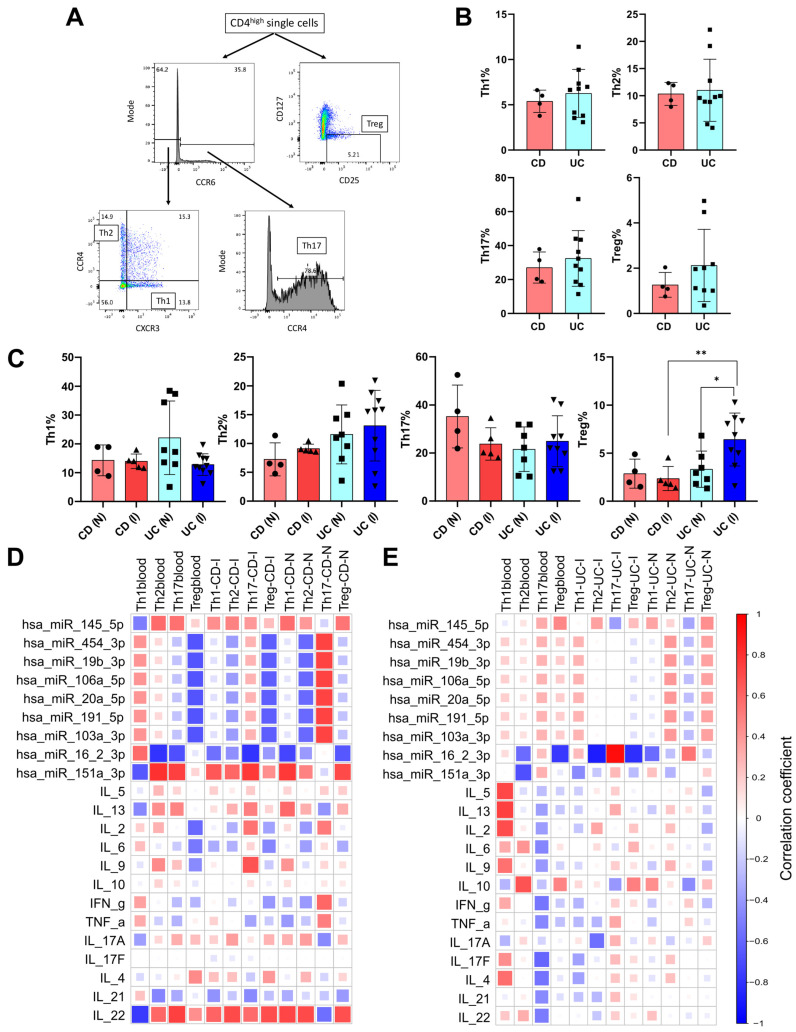
Characterization and correlations of circulating and tissue CD4^+^ T cell subset distribution in CD and UC patients. (**A**) Representative flow cytometry plots to illustrate the gating strategy for Fluorescence-Activated Cell Sorting (FACS) analysis of T cell subsets. (**B**) The percentage of T helper 1 (Th1), T helper 2 (Th2), T helper 17 (Th17), and regulatory T (Treg) cells in the circulation as determined by flow cytometry in patients with CD and UC. (**C**) The distribution of different T cell subsets measured in intestinal biopsy samples taken from inflamed and non-inflamed spots of CD and UC patients. For flow cytometry, we analyzed the blood and tissue samples from 4 to 5 CD and 8 to 10 UC patients. Data are shown as individual values and mean ± SD expressed as the percentage of total CD4^+^ cells of the given sample. Asterisks report statistical significance * *p* < 0.05; ** *p* < 0.005. Statistical analyses between groups were performed using the Kruskal–Wallis test with Dunn’s multiple-comparison. Spearman correlation coefficients were calculated to describe the correlation between T cell subset frequencies in blood and biopsy samples, as well as levels of circulating miRNAs in (**D**) CD and (**E**) UC patients. The correlation coefficients vary from −1 to +1, and the color coding indicates the strength of positive (R ≥ 0; red) or negative (R < 0; blue) correlation. CD (Crohn disease), UC (Ulcerative colitis), CD (I) (inflamed tissue biopsy sample from CD), CD (N) (non-inflamed tissue biopsy sample from CD), UC (I) (inflamed tissue biopsy sample from UC), UC(N) (non-inflamed tissue biopsy sample from UC).

**Figure 6 ijms-26-12042-f006:**
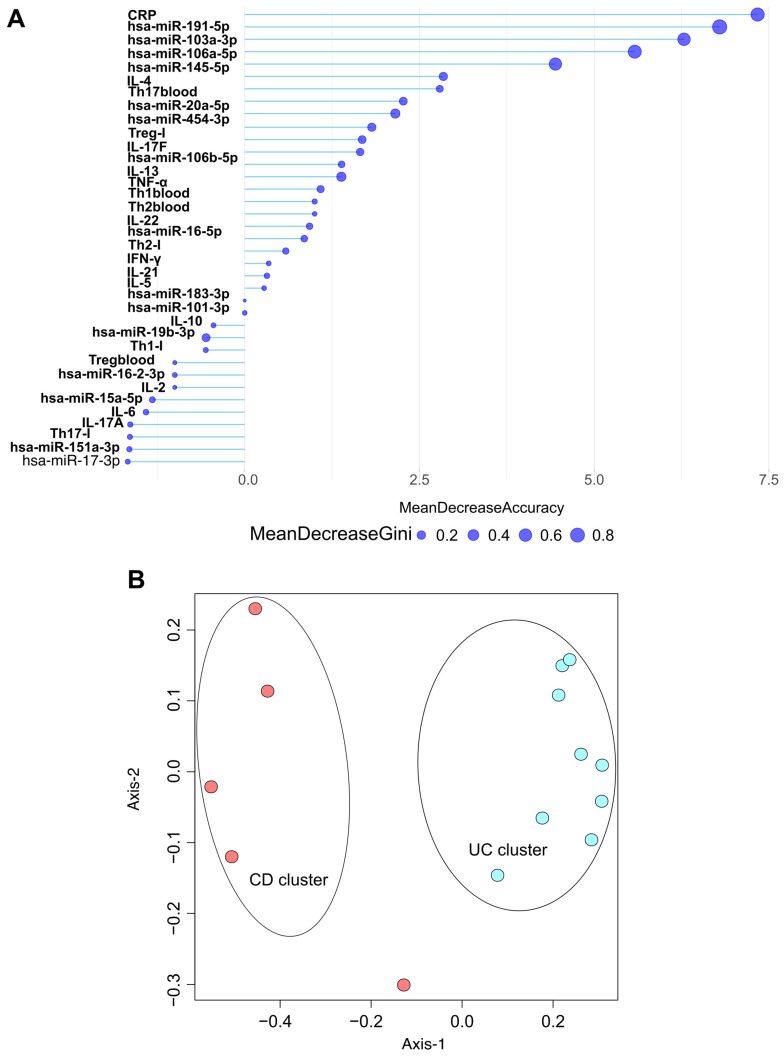
Random Forest model discriminates CD and UC patients based on biomarkers. (**A**) Partial dependence plot represents the variable importance of normalized biomarker levels, including C-reactive protein (CRP), differentially expressed microRNAs, cytokine levels, and CD4^+^ T cell subset distributions in blood and inflamed tissue according to mean decrease in Gini and mean decrease in accuracy coefficients. (**B**) Multidimensional scaling (MDS) plot of the Random Forest proximity matrix reveals distinct clusters of CD and UC patient groups. Each dot displays a patient (red circles: CD patients; blue circles: UC patients). The spatial distance between points reflects the dissimilarity between samples, with greater distance indicating more pronounced differences. CD (Crohn’s disease), UC (Ulcerative colitis). I (inflamed biopsy sample).

**Table 1 ijms-26-12042-t001:** The baseline characteristics and clinical parameters of the enrolled patients are presented according to the study groups. The data are shown as mean ± standard deviation (SD) and percentage (%). UC: Ulcerative Colitis, CD: Crohn’s Disease, CDAI: (Adult) Crohn’s Disease Activity Index, eMAYO: (Adult) Disease Activity Index for Ulcerative Colitis, PCDAI: Pediatric Crohn’s Disease Activity Index, PUCAI: Pediatric Ulcerative Colitis Activity Index, WBC: White Blood Cells, HGB: Hemoglobin, CRP: C-Reactive Protein.

Characteristics	Patients with Ulcerative Colitis (UC) (*n* = 10)	Patients with Crohn’s Disease (CD) (*n* = 8)	Healthy Control (*n* = 9)
**Demographics**			
Age (years ± SD)	31.1 (±5.85)	24.5 (±10.8)	30.9 (±5.67)
Male, n (%)	6 (60%)	4 (50%)	5 (55.6%)
Female, n (%)	4 (40%)	4 (50%)	4 (44.4%)
**Illness severity score (** **±SD)**			
Colitis Clinical Score (CDAI)	232.2 (±89.15)		
Crohn Clinical Score (eMAYO)		2 (0)	
PCDAI		23.33 (±2.88)	
PUCAI	20 (0)		
**Comorbidities**			
Diarrhea every day	9 (90%)	4 (50%)	
Diarrhea (n/day ± SD)	3.80 (±5.65)	3.12 (±3.83)	
**Laboratory parameters (** **±SD)**			
WBC (g/L)	7.04 (±2.50)	11.08 (±5.73)	
HGB (g/L)	141.17 (±25.71)	120.13 (±27.35)	
Neutrophils (%)	64.65 (±10.56)	72.25 (±7.09)	
Lymphocytes (%)	24.40 (±9.33)	14.64 (±5.83)	
Monocytes (%)	7.72 (±2.17)	9.15 (±4.50)	
Eosinophils (%)	0.80 (±0.38)	2.91 (±2.71)	
Basophils (%)	0.30 (±0.18)	0.23 (±0.18)	
CRP (mg/L)	3.74 (±3.12)	62.44 (±71.19)	

**Table 2 ijms-26-12042-t002:** Unique miRNAs identified between the following groups: CD and UC; UC and control; CD and control. CD (Crohn’s disease), UC (Ulcerative colitis).

CD and UC Groups	UC and Control Groups	CD and Control Groups
hsa-miR-10395-3p	hsa-miR-6837-3p	hsa-miR-6509-3p
hsa-miR-1248	hsa-miR-1246	
hsa-miR-195-3p	hsa-miR-374c-5p	

## Data Availability

The original data presented in the study are openly available in [NCBI Sequence Read Archive (SRA)] at [https://www.ncbi.nlm.nih.gov/sra/PRJNA1302731] (accessed on 7 August 2025).

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
