# Peer review of "Interplay Between Dysregulated Immune System and the Footprints of Blood-Borne miRNAs in Treatment Naive Crohn’s Disease and Ulcerative Colitis Patients"

_ijms, 2025, doi:10.3390/ijms262412042_

Round 1

Reviewer 1 Report

Comments and Suggestions for Authors

Thank you for submitting your manuscript titled “Interplay between dysregulated immune system and the footprints of blood-borne miRNAs in treatment naïve Crohn’s disease and ulcerative colitis patients.” This is a promising and timely study that explores the potential of miRNA signatures as non-invasive biomarkers for distinguishing between CD and UC. The integrative approach—combining small RNA-seq, cytokine profiling, and flow cytometric immune cell characterization—is commendable.

However, several major issues need to be addressed before the manuscript can be considered suitable for publication in a high-impact journal:

1. Novelty and Literature Context

While the focus on therapy-naïve patients adds rigor and clinical relevance, the novelty of the identified miRNAs is limited. Most of the validated miRNAs (e.g., miR-106a, miR-145, miR-20a) have previously been associated with IBD and/or colorectal cancer. A more in-depth and critical comparison to recent literature is needed. Additionally, some important miRNAs (e.g., miR-155, miR-223) and key IBD-related regulatory mechanisms (e.g., JAK-STAT, microbiota interactions) are missing from the discussion.

Recommendation: Expand the background and discussion to better position your findings within the current state of knowledge. Highlight what is truly novel and address any gaps in prior citations.

2. Methodology Clarification

The methodology section is detailed, but further clarification is needed in several areas:

  • RNA-seq processing steps (e.g., normalization, batch correction, tools used)

  • Patient stratification (e.g., adult vs pediatric subgroups, disease extent)

  • Criteria for ROC analysis (e.g., cut-offs, cross-validation method)

Recommendation: Clearly state statistical corrections for multiple testing (e.g., FDR), address sample size limitations, and provide effect sizes or confidence intervals where possible.

3. Sample Size and Statistical Power

The small cohort size (CD: 8, UC: 10, Control: 9) significantly limits the generalizability and statistical robustness of your findings. While the inclusion of therapy-naïve patients is appreciated, the results—particularly the diagnostic potential of miRNAs—should be interpreted more cautiously.

Recommendation: Temper claims about diagnostic utility and emphasize the exploratory nature of this study. Consider this as a discovery phase needing further validation.

4. Interpretation of Findings

The manuscript occasionally overstates the clinical implications of findings. While the identified miRNAs may be promising biomarkers, their diagnostic and prognostic potential requires validation in larger, independent cohorts.

Recommendation: Reframe sections that imply direct clinical utility (e.g., “diagnostic value,” “may guide therapeutic decisions”) unless supported by external validation.

5. Figures, Tables & Supplementary Materials

Some figures—especially complex correlation heatmaps and random forest plots—are difficult to interpret due to excessive detail and insufficiently clear legends. The presentation of ROC curves and clustering could also be simplified for clarity.

Recommendation: Improve figure legends and consider breaking down complex visuals. Clearly label patient groups and use color coding consistently. Provide per-patient qPCR values in supplementary material.

6. Language and Writing Style

There are consistent issues with grammar, phrasing, and clarity throughout the manuscript. Phrases such as “miRNAs eventuated increased expression” should be revised for correctness and clarity. Passive voice is overused, and some sentences are overly long or redundant.

Recommendation: Substantially revise the manuscript for English language clarity. A professional scientific editing service may be useful.

7. Suggestions for Strengthening the Manuscript

  • Discuss potential mechanisms linking the five validated miRNAs to T-cell regulation and IBD pathogenesis.

  • Consider conducting pathway enrichment analysis of miRNA targets to provide functional context.

  • If possible, include reference to ongoing or future validation efforts.

  • Reassess whether some conclusions drawn from correlation networks are overinterpreted without mechanistic validation.

Comments on the Quality of English Language

This manuscript titled “Interplay between dysregulated immune system and the footprints of blood-borne miRNAs in treatment naïve Crohn’s disease and ulcerative colitis patients” presents an integrative investigation of miRNA expression, T-cell subset profiling, and cytokine signatures in treatment-naïve IBD patients. The study aims to explore the diagnostic potential of circulating miRNAs in distinguishing Crohn’s Disease (CD) from Ulcerative Colitis (UC).

Strengths:

  • The authors use a multi-modal approach combining small RNA-sequencing, RT-qPCR validation, cytokine profiling, and immune phenotyping.

  • The patient cohort is strictly therapy-naïve, minimizing confounding variables from medication.

  • Five candidate miRNAs (e.g., miR-106a-5p, miR-145-5p) showed differential expression between CD and UC, with excellent ROC metrics in this cohort.

Concerns:

  • The sample size is very limited (n=8 CD, n=10 UC, n=9 controls), which restricts statistical power and undermines the robustness of diagnostic claims.

  • Most of the validated miRNAs are not novel to the field of IBD biomarker research, and the manuscript does not clearly position its contribution against recent studies.

  • Several key methodological details (e.g., RNA-seq normalization, statistical corrections) require clarification.

  • The language and structure of the manuscript need substantial revision for clarity and conciseness.

Recommendation:

Although the study is thoughtfully designed and tackles a meaningful clinical question, the manuscript requires major revisions before it can be considered suitable for publication. The authors should revise the manuscript to:

  • Improve clarity of writing and figures,

  • Provide stronger comparisons to existing literature,

  • Acknowledge the exploratory nature of their diagnostic findings due to small sample size,

  • Enhance methodological transparency.

Author Response

Thank you for submitting your manuscript titled “Interplay between dysregulated immune system and the footprints of blood-borne miRNAs in treatment naïve Crohn’s disease and ulcerative colitis patients.” This is a promising and timely study that explores the potential of miRNA signatures as non-invasive biomarkers for distinguishing between CD and UC. The integrative approach—combining small RNA-seq, cytokine profiling, and flow cytometric immune cell characterization—is commendable.

However, several major issues need to be addressed before the manuscript can be considered suitable for publication in a high-impact journal:

The authors are grateful for the thorough review as well as for the constructive comments and suggestions regarding the manuscript. We implemented a substantial revision of the whole manuscript to improve language quality. Additionally, we tried to address all comments and suggestions raised by the reviewer according to our detailed point-by-point response. Given the amount of newly added and modified text content of the manuscript, the changes compared to the original submission are not highlighted. We hope that this does not cause any inconvenience to the reviewer.

  1. Novelty and Literature Context

While the focus on therapy-naïve patients adds rigor and clinical relevance, the novelty of the identified miRNAs is limited. Most of the validated miRNAs (e.g., miR-106a, miR-145, miR-20a) have previously been associated with IBD and/or colorectal cancer. A more in-depth and critical comparison to recent literature is needed. Additionally, some important miRNAs (e.g., miR-155, miR-223) and key IBD-related regulatory mechanisms (e.g., JAK-STAT, microbiota interactions) are missing from the discussion.

Recommendation: Expand the background and discussion to better position your findings within the current state of knowledge. Highlight what is truly novel and address any gaps in prior citations.

Thank you for this insightful comment. We agree that several of the validated miRNAs identified in our cohort (including miR-106a, miR-145, and miR-20a) have been previously described to be associated with IBD and colorectal cancer. In the revised manuscript, we have substantially expanded and rephrased the introduction and discussion parts to provide more detailed comparison with recent studies, highlighting both the consistent and the unique aspects of our findings in therapy-naïve patients.

  1. Methodology Clarification

The methodology section is detailed, but further clarification is needed in several areas:

  • RNA-seq processing steps (e.g., normalization, batch correction, tools used)
  • Patient stratification (e.g., adult vs pediatric subgroups, disease extent)
  • Criteria for ROC analysis (e.g., cut-offs, cross-validation method)

Recommendation: Clearly state statistical corrections for multiple testing (e.g., FDR), address sample size limitations, and provide effect sizes or confidence intervals where possible.

We thank the reviewer for these constructive comments. The Methods section has now been clarified and expanded: We added details explaining the lack of further patient stratification and subgrouping. Additionally, we have added a step-by-step description of the miRNA-seq pre-processing workflow, including Cutadapt filtering, FastQC quality checks, Trimmomatic trimming, and miRge 2.0-based miRNA annotation.

Also, normalization and differential expression analysis procedures are now explicitly described, including TMM normalization in edgeR and filtering criteria reporting. We added a detailed explanation of the ROC analysis, including the use of easyROC, AUC estimation, confidence intervals, and cut-off determination using Youden’s index. We also added detailed information about the software packages used in bioinformatics analyses. We have amended and corrected the statistical analyses part as well.

  1. Sample Size and Statistical Power

The small cohort size (CD: 8, UC: 10, Control: 9) significantly limits the generalizability and statistical robustness of your findings. While the inclusion of therapy-naïve patients is appreciated, the results—particularly the diagnostic potential of miRNAs—should be interpreted more cautiously.

Recommendation: Temper claims about diagnostic utility and emphasize the exploratory nature of this study. Consider this as a discovery phase needing further validation.

The results and discussion parts have been modified as requested. We emphasized the exploratory nature of our study and strongly highlighted that further validations are needed to draw far-reaching conclusions. Furthermore, we have moved the ROC analyses from the main Figure 3 to Supplementary Figure 1 as we agree with the reviewer, that due to the relatively small sample size, the direct applicability of such an approach is highly limited.

  1. Interpretation of Findings

The manuscript occasionally overstates the clinical implications of findings. While the identified miRNAs may be promising biomarkers, their diagnostic and prognostic potential requires validation in larger, independent cohorts.

Recommendation: Reframe sections that imply direct clinical utility (e.g., “diagnostic value,” “may guide therapeutic decisions”) unless supported by external validation.

We have carefully revised the manuscript to avoid overstating the clinical implications of our findings. Statements with direct diagnostic or therapeutic relevance have been reframed to emphasize that the identified miRNAs represent preliminary candidates rather than validated biomarkers. As mentioned before, we have also moved the ROC analyses from the main figures a supplementary figure, to avoid overstating our conclusions.

  1. Figures, Tables & Supplementary Materials

Some figures—especially complex correlation heatmaps and random forest plots—are difficult to interpret due to excessive detail and insufficiently clear legends. The presentation of ROC curves and clustering could also be simplified for clarity.

Recommendation: Improve figure legends and consider breaking down complex visuals. Clearly label patient groups and use color coding consistently. Provide per-patient qPCR values in supplementary material.

We modified the complex heatmap shown in Fig 2D and we also modified Fig 3: the ROC analyses have moved to the supplementary Fig 1. Figure legends were also revised to improve clarity. Additionally, we provided the raw, per-patient Ct values of our RT-qPCR validation in supplementary table 5.

  1. Language and Writing Style

There are consistent issues with grammar, phrasing, and clarity throughout the manuscript. Phrases such as “miRNAs eventuated increased expression” should be revised for correctness and clarity. Passive voice is overused, and some sentences are overly long or redundant.

Recommendation: Substantially revise the manuscript for English language clarity. A professional scientific editing service may be useful.

Thank you for this very important comment. We have thoroughly revised the whole manuscript for grammar and spelling mistakes and rephrased the whole text. On top of this, we substantially expanded the “Introduction” and “Discussion” chapters. We believe that our efforts improved language- as well as scientific clarity of the manuscript

  1. Suggestions for Strengthening the Manuscript
  • Discuss potential mechanisms linking the five validated miRNAs to T-cell regulation and IBD pathogenesis.
  • Consider conducting pathway enrichment analysis of miRNA targets to provide functional context.
  • If possible, include reference to ongoing or future validation efforts.
  • Reassess whether some conclusions drawn from correlation networks are overinterpreted without mechanistic validation.

The authors are grateful for the reviewer’s suggestions to strengthen the manuscript. We have expanded our discussion to put our results deeper into scientific context, that also relates our five validated miRNAs to existing literature. We agree, that providing functional context is of great importance, however, as highlighted by the reviewers, our study is rather exploratory, therefore we believe that a pathway enrichment analysis would only confer relevant insights using a bigger, stratified dataset. We also agree that our exploratory data requires further internal or external validation, therefore we have also tempered the claims of our current manuscript.

Comments on the Quality of English Language

This manuscript titled “Interplay between dysregulated immune system and the footprints of blood-borne miRNAs in treatment naïve Crohn’s disease and ulcerative colitis patients” presents an integrative investigation of miRNA expression, T-cell subset profiling, and cytokine signatures in treatment-naïve IBD patients. The study aims to explore the diagnostic potential of circulating miRNAs in distinguishing Crohn’s Disease (CD) from Ulcerative Colitis (UC).

Strengths:

  • The authors use a multi-modal approach combining small RNA-sequencing, RT-qPCR validation, cytokine profiling, and immune phenotyping.
  • The patient cohort is strictly therapy-naïve, minimizing confounding variables from medication.
  • Five candidate miRNAs (e.g., miR-106a-5p, miR-145-5p) showed differential expression between CD and UC, with excellent ROC metrics in this cohort.

Concerns:

  • The sample size is very limited (n=8 CD, n=10 UC, n=9 controls), which restricts statistical power and undermines the robustness of diagnostic claims.
  • Most of the validated miRNAs are not novel to the field of IBD biomarker research, and the manuscript does not clearly position its contribution against recent studies.
  • Several key methodological details (e.g., RNA-seq normalization, statistical corrections) require clarification.
  • The language and structure of the manuscript need substantial revision for clarity and conciseness.

Recommendation:

Although the study is thoughtfully designed and tackles a meaningful clinical question, the manuscript requires major revisions before it can be considered suitable for publication. The authors should revise the manuscript to:

  • Improve clarity of writing and figures,
  • Provide stronger comparisons to existing literature,
  • Acknowledge the exploratory nature of their diagnostic findings due to small sample size,
  • Enhance methodological transparency.

As detailed in our point-by-point response above, we believe that we have addressed the reviewer’s recommendations by improving language and figure clarity, proper in-depth methodological description and also by providing deeper scientific context to our findings, as well as by tempering our claims and highlighting the exploratory nature of our study.

Reviewer 2 Report

Comments and Suggestions for Authors

Comments,

In this manuscript, the authors investigated the relationship between dysregulated immune responses and altered miRNAs in patients with inflammatory bowel disease. First, miRNA sequencing of healthy volunteers and IBD patients was screened, and 14 miRNAs were identified with significant alterations. Furthermore, circulating T-cell-related cytokines and T-cell subpopulations in healthy volunteers, Crohn’s disease patients, and ulcerative colitis patients were measured to investigate these correlations. Overall, the manuscript is poorly written, some figures contain errors, and the descriptions for some results were incorrect. The issues are listed below.

  1. The diagrams in Fig. 1B only summarized the quantities of miRNAs with high, medium, and low PRM values but failed to convey the alterations in individual miRNAs, which is critical for the reader. The reviewer suggests the authors provide detailed data for individual miRNAs.
  2. “log104,” “log102,” and “log101” in Fig. 1 should be corrected to log104, log102, and log101.
  3. It would be valuable for the authors to discuss the potential functions of the unique miRNAs presented in Table 2, as they are not explored in the later sections of the manuscript.
  4. In Fig. 2D, the authors should present the miRNA alterations in distinct, group-specific clusters for the control, CD, and UC cohorts.
  5. The sequence of primers for the RT-qPCR should be listed.
  6. Based on the results in Section 2.5, five specific miRNAs were selected. The authors should describe the functions of all these miRNAs in the Discussion section, rather than focusing solely on miR-101-3p.
  7. In Fig. 4C, IL-21 has a positive correlation with most of these microRNAs in the UC patient group. Is the description in Section 2.6 correct?
  8. The names of microRNAs in the article should be consistent.
  9. In Fig. 4C, the correlation level of IL-10 with hsa-mir-15a-5p is similar. Is the description in Section 2.6 correct?
  10. In section 2.6 the sentence “…, whereas in CD only weak correlations were observed” is incorrect. Authors should use “UC” to replace “CD.”

Author Response

In this manuscript, the authors investigated the relationship between dysregulated immune responses and altered miRNAs in patients with inflammatory bowel disease. First, miRNA sequencing of healthy volunteers and IBD patients was screened, and 14 miRNAs were identified with significant alterations. Furthermore, circulating T-cell-related cytokines and T-cell subpopulations in healthy volunteers, Crohn’s disease patients, and ulcerative colitis patients were measured to investigate these correlations. Overall, the manuscript is poorly written, some figures contain errors, and the descriptions for some results were incorrect. The issues are listed below.

The authors are very grateful for the reviewer’s comments and suggestions regarding our manuscript. We implemented a thorough revision of the whole manuscript to improve language quality. Additionally, we tried to address all comments and suggestions raised by the reviewer according to our detailed point-by-point response.

  1. The diagrams in Fig. 1B only summarized the quantities of miRNAs with high, medium, and low RPM values but failed to convey the alterations in individual miRNAs, which is critical for the reader. The reviewer suggests the authors provide detailed data for individual miRNAs.

To address this comment, we have now provided a detailed table (new supplementary table 2.) listing all the RPM values of all miRNAs with RPM > 10 across all samples in addition to the summary diagrams shown in Fig. 1B.

  1. “log104,” “log102,” and “log101” in Fig. 1 should be corrected to log104, log102, and log101.

Figure 1. was corrected as requested.

  1. It would be valuable for the authors to discuss the potential functions of the unique miRNAs presented in Table 2, as they are not explored in the later sections of the manuscript.

Thank you for this important note, we have now expanded our “Discussion” and included a short overview about the described biological functions of the miRNAs listed in Table 2.

  1. In Fig. 2D, the authors should present the miRNA alterations in distinct, group-specific clusters for the control, CD, and UC cohorts.

We have modified the figure 2D, as requested by the reviewer.

  1. The sequence of primers for the RT-qPCR should be listed.

The list of RT-qPCR primers used for analyses are listed in Supplementary Table 4 as requested.

  1. Based on the results in Section 2.5, five specific miRNAs were selected. The authors should describe the functions of all these miRNAs in the Discussion section, rather than focusing solely on miR-101-3p.

We have expanded our “Discussion” chapter to include descriptions about the known or suspected biological functions of all five miRNAs listed in Section 2.5

  1. In Fig. 4C, IL-21 has a positive correlation with most of these microRNAs in the UC patient group. Is the description in Section 2.6 correct?

We appreciate the reviewer’s careful observation. The description in Section 2.6 was indeed incorrect. We modified the description to “Only weak positive correlations were observed between IL-21 and microRNAs in UC”.

  1. The names of microRNAs in the article should be consistent.

We have carefully reviewed the entire manuscript and standardized all miRNA names according to the miRBase convention. This correction has been applied consistently across the text, figures, and tables.

  1. In Fig. 4C, the correlation level of IL-10 with hsa-mir-15a-5p is similar. Is the description in Section 2.6 correct?

We have corrected the sentence to “In contrast, only weak positive correlation was observed between IL-10 and hsa-mir-15a-5p (r=0.43) in CD”.

  1. In section 2.6 the sentence “…, whereas in CD only weak correlations were observed” is incorrect. Authors should use “UC” to replace “CD.”

We have corrected the noted and indeed false sentence; CD is now changed to UC in that sentence.

Round 2

Reviewer 1 Report

Comments and Suggestions for Authors

The manuscript investigates the role of blood-borne miRNAs in immune dysregulation in Crohn’s disease (CD) and ulcerative colitis (UC), aiming to identify diagnostic biomarkers for these diseases. The study is well-designed, using advanced techniques such as small RNA sequencing and cytokine profiling. The results highlight significant differences in miRNA expression between IBD patients and healthy controls, and the authors effectively use random forest and multidimensional scaling (MDS) to differentiate patient groups.

However, the manuscript could benefit from a more focused introduction, clearly differentiating the novelty of this study from previous research. While the methods are solid, further justification for miRNA selection and more detail on sample size calculations would enhance clarity. The results section is well-structured, but more discussion on the functional roles of the identified miRNAs in disease pathogenesis would improve the depth of the study. Although the figures and tables support the analysis, clearer explanations of the MDS plots and figure legends are needed. The manuscript's language is generally clear but could be simplified in some areas for better accessibility.

In the discussion, the authors could further elaborate on the potential clinical applications of their findings, such as diagnostic or therapeutic strategies based on miRNA profiles. Finally, while the references are mostly relevant, including a few additional key studies would strengthen the manuscript’s foundation.

Author Response

The authors are thankful for the further comments of the reviewer. We addressed the remaining concerns as follows: 

Comments and Suggestions for Authors

The manuscript investigates the role of blood-borne miRNAs in immune dysregulation in Crohn’s disease (CD) and ulcerative colitis (UC), aiming to identify diagnostic biomarkers for these diseases. The study is well-designed, using advanced techniques such as small RNA sequencing and cytokine profiling. The results highlight significant differences in miRNA expression between IBD patients and healthy controls, and the authors effectively use random forest and multidimensional scaling (MDS) to differentiate patient groups.

1) However, the manuscript could benefit from a more focused introduction, clearly differentiating the novelty of this study from previous research.

In response, we have revised, restructured, and amended our “Introduction” to more clearly articulate the novelty of our study and explicitly differentiate it from previous research. This integrative, systems-level approach enables the identification of treatment-independent, early molecular alterations and allows us to delineate disease-specific miRNA–cytokine–T-cell interaction networks that differentiate CD from UC. Notably, we also map the validated miRNA panel onto pathways implicated in colitis-associated colorectal cancer, thereby positioning the miRNA–cytokine–T-cell axis as a mechanistically informative and potentially clinically actionable biomarker framework. These points have now been emphasized more clearly in the revised Introduction (New text is highlighted in yellow).

2) While the methods are solid, further justification for miRNA selection and more detail on sample size calculations would enhance clarity.

We corrected the results “2.4. Differentially expressed miRNAs were revealed between IBD patients and healthy groups” parts. We added the following extra sentences to further justify miRNA selection. “For PCR validation, we selected 62 miRNAs (Figure 2D) based on our small RNA sequencing results and previously published studies relevant to intestinal biology”. Furthermore, we added additional details to the Materials and Methods section to explain the rationale behind the selection of the miRNAs. “Based on the results of small RNA sequencing, we selected a subset of 62 miRNAs for further validation. Selection criteria included the magnitude of differential expression between IBD patients and healthy controls, as well as prior evidence from the literature implicating these miRNAs in intestinal physiology, immune regulation, or inflammatory bowel disease pathogenesis.” We hope that, together with the references already included, this provides sufficient justification for our miRNA selection.

3) The results section is well-structured, but more discussion on the functional roles of the identified miRNAs in disease pathogenesis would improve the depth of the study.

We added some amendments to our text to provide more information about the functional roles of the identified miRNAs as requested. (New text is highlighted in yellow).

4) Although the figures and tables support the analysis, clearer explanations of the MDS plots and figure legends are needed.

We thoroughly revised and corrected the figure legends and added a more detailed description of MDS plots.

5) The manuscript's language is generally clear but could be simplified in some areas for better accessibility.

The wording and compound expressions were simplified and streamlined throughout the text, especially focusing on improving clarity of the “Introduction” and “Discussion” chapters

6) In the discussion, the authors could further elaborate on the potential clinical applications of their findings, such as diagnostic or therapeutic strategies based on miRNA profiles.

We added the following extra sentences to further elaborate on the clinical application potential based on miRNA profiles. “Our preliminary data suggest that hsa-miR-106a-5p, hsa-miR-103a-3p, hsa-miR-191-5p, hsa-miR-145-5p, and hsa-miR-20a-5p may serve as potential biomarkers for IBD, with the ability to discriminate ulcerative colitis from Crohn’s disease. Nevertheless, larger cohorts and additional validation analyses will be required to confirm their diagnostic utility.” We believe that adding this piece of information should be sufficient to support the recommendation of the reviewer along with keeping our conclusions and the applicability of our exploratory observations moderate enough.

7) Finally, while the references are mostly relevant, including a few additional key studies would strengthen the manuscript’s foundation.

The following key studies were further referenced in our manuscript, to strengthen the scientific context and ensure a more comprehensive foundation for our work.

  1. Amado, T.; Schmolka, N.; Metwally, H.; Silva-Santos, B.; Gomes, A.Q. Cross-Regulation between Cytokine and microRNA Pathways in T Cells. Eur J Immunol 2015, 45, 1584–1595, doi:10.1002/eji.201545487.
  2. Asangani, I.A.; Rasheed, S. a. K.; Nikolova, D.A.; Leupold, J.H.; Colburn, N.H.; Post, S.; Allgayer, H. MicroRNA-21 (miR-21) Post-Transcriptionally Downregulates Tumor Suppressor Pdcd4 and Stimulates Invasion, Intravasation and Metastasis in Colorectal Cancer. Oncogene 2008, 27, 2128–2136, doi:10.1038/sj.onc.1210856.
  3. Pathak, S.; Grillo, A.R.; Scarpa, M.; Brun, P.; D’Incà, R.; Nai, L.; Banerjee, A.; Cavallo, D.; Barzon, L.; Palù, G.; et al. MiR-155 Modulates the Inflammatory Phenotype of Intestinal Myofibroblasts by Targeting SOCS1 in Ulcerative Colitis. Exp Mol Med 2015, 47, e164, doi:10.1038/emm.2015.21.
  4. Mozammel, N.; Amini, M.; Baradaran, B.; Mahdavi, S.Z.B.; Hosseini, S.S.; Mokhtarzadeh, A. The Function of miR-145 in Colorectal Cancer Progression; an Updated Review on Related Signaling Pathways. Pathol Res Pract 2023, 242, 154290, doi:10.1016/j.prp.2022.154290.
  5. Zhao, X.; Cui, D.; Yuan, W.; Chen, C.; Liu, Q. Berberine Represses Wnt/β-Catenin Pathway Activation via Modulating the microRNA-103a-3p/Bromodomain-Containing Protein 4 Axis, Thereby Refraining Pyroptosis and Reducing the Intestinal Mucosal Barrier Defect Induced via Colitis. Bioengineered 2022, 13, 7392–7409, doi:10.1080/21655979.2022.2047405.
  6. Majd, M.; Hosseini, A.; Ghaedi, K.; Kiani-Esfahani, A.; Tanhaei, S.; Shiralian-Esfahani, H.; Rahnamaee, S.Y.; Mowla, S.J.; Nasr-Esfahani, M.H. MiR-9-5p and miR-106a-5p Dysregulated in CD4+ T-Cells of Multiple Sclerosis Patients and Targeted Essential Factors of T Helper17/Regulatory T-Cells Differentiation. Iran J Basic Med Sci 2018, 21, 277–283, doi:10.22038/ijbms.2018.25382.6275.

Reviewer 2 Report

Comments and Suggestions for Authors

No further comments.

Author Response

The authors are grateful for the previous constructive recommendations of the reviewer.